# AI-assisted optimization design of seismic performance parameters for timber structures

**Dongqi Wei[1], Yuqiang Ding[1], Feng Zhou[1], Xuan Zhang[2]\***

**1** School of Architectural Engineering and Art Design, Liuzhou City Vocational College, Liuzhou, Guangxi, China, **2** School of Environmental and Life Sciences, Nanning Normal University, Nanning, Guangxi, China

\* zhangxuan1193@outlook.com

## Abstract

Timber multi-story buildings offer environmental benefits, lightweight construction, and seismic resilience, but Artificial Intelligence (AI) based integrated frameworks for optimizing seismic parameters, including inter-story drift and roof displacement, remain limited. The Gradient Boosted Random Forest Machine with Scalable Cheetah Optimizer (GBRF-SCO) is proposed for improving prediction accuracy while facilitating optimal design decisions. The dataset consists of 4,000 timber building samples obtained from a publicly available Kaggle repository (Timber Seismic Performance Dataset). Data pre-processing employs normalization and outlier detection using Robust Scaling and Isolation Forest, ensuring high-quality inputs. For exploratory analysis, t-Distributed Stochastic Neighbor Embedding (t-SNE) is applied to visualize high-dimensional feature relationships and identify structural parameter patterns relevant to seismic performance. The proposed framework uses GBRF to predict seismic response metrics, with the SCO tweaking hyperparameters to optimize model performance. It also enables the optimization of seismic performance characteristics, guiding engineers in selecting structural designs that minimize drift and enhance robustness. Multiple Linear Regression (MLR) was employed to examine the influence of key structural and seismic elements on roof displacement, providing insights into the overall seismic performance of wood buildings. Comparative evaluation shows superior performance over conventional regression and ensemble methods, demonstrating a higher accuracy of 0.949, which corresponds to the classification of roof displacement levels (low, medium, high) under seismic loading conditions and seismic intensities using Python 3.10. By providing a strong and clever method for designing sustainable and earthquake-resilient buildings, the suggested GBRF-SCO framework successfully improves the seismic performance optimization of timber structures.

**Data availability statement:** All relevant data are within the manuscript.

**Funding:** The author(s) received no specific funding for this work.

**Competing interests:** The authors have declared that no competing interests exist.

## 1. Introduction

Wood constructions play a vital role in modern building due to their exclusive grouping of strength, flexibility, and sustainability, promoting both builders and landlords of large-scale infrastructures [1,2]. Over the past 50 years, loans in engineering techniques have enabled wood to transition from a traditional framing material to a primary structural component in commercial and residential buildings. Engineered timber products enhance strength, stability, and fire resistance compared to conventional wood-framed structures, while offering a higher strength-to-weight ratio that facilitates faster construction and reduces environmental impact relative to concrete or steel [3,4]. Technological developments ensure timber remains a durable, adaptable, and sustainable material choice in contemporary design.

Earthquake performance is a critical consideration for timber buildings, as seismic forces can induce inter-story drift, roof displacement, base shear, acceleration, and energy dissipation [5,6]. Evaluating these performance indicators allows engineers to determine whether a structure possesses sufficient strength, stability, and energy-dissipation capacity to resist seismic events [7]. Key metrics include lateral displacement, inter-story drift, base shear, ductility, damping capacity, and the natural period of vibration, which collectively describe a timber building's ability to withstand seismic loads without catastrophic failure. Adherence to earthquake-resistant design principles ensures human safety and structural resilience [8].

Engineers improve seismic performance by modifying structural systems, material selection, and construction details, while verifying that buildings are safe, serviceable, and compliant with codes [9,10]. Specific seismic performance parameters, essential for assessing timber structures, rely on intrinsic wood properties such as ductility, low mass, and flexibility [11,12]. Specifically, lateral displacement, inter-story drift, base shear, damping capacity, and energy dissipation are key indicators of timber structure resilience under seismic loading [13]. Moreover, the form, layout, and detailing of timber connections, as well as the use of engineered wood products, significantly influence seismic performance [14]. Proper assessment ensures timber structures can safely absorb and transfer seismic energy, limiting damage and protecting occupants. Conforming to current seismic codes and performance-based design standards requires careful consideration of these factors throughout the design and construction process [15]. Fig 1 shows the detailed structural framework of a timber building.

It is crucial to monitor seismic performance, such as lateral drifts and roof displacement, to ensure that structures are safe under earthquake loads.

The research introduces the GBRF-SCO framework, an AI-driven methodology that enhances essential seismic metrics, including inter-story drift and roof displacement. Engineers can use the model's hyperparameter optimization to make better predictions and design buildings that are more resistant to earthquakes and less likely to drift. In general, the GBRF-SCO framework helps with safer, stronger, and faster design of timber buildings.

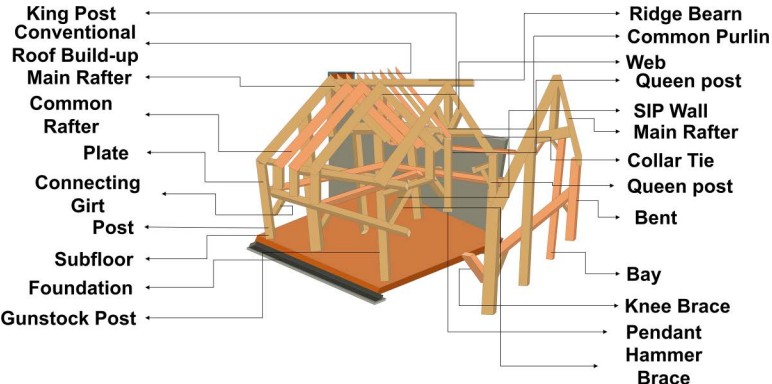

**Fig 1. Structural framework of a timber building illustrating key components influencing seismic performance.** [Timber-Frames: Anatomy and Joinery – Custom Home Building and Remodeling].

### 1.1. Key contribution of the research

- Initially, analytical modeling was carried out using a comprehensive Timber Seismic Performance Dataset that included different seismic and structural characteristics such as height, density, and acceleration.

- After collecting the data, the dataset was processed through robust scaling, isolation forest, outlier removal, and normalization to enhance the model's accuracy and reliability.

- t-SNE analysis to visualize relationships between structural and seismic variables, identifying key trends for performance optimization.

- Demonstration that the GBRF-SCO framework achieves superior predictive performance and stability, with improved accuracy, precision, and F1-score over conventional models.

## 2. Literature review

### 2.1. AI and machine learning in timber seismic performance

The research aimed to develop a multi-objective optimization framework [16] leveraging Deep Learning (DL) for the design of 20-story cross-laminated timber (CLT) Coupled Wall systems, improving high-dimensional design handling. An autoencoder compressed the design space dimensions, while neural networks mapped input variables to latent spaces and linked latent variables to output responses for structural optimization. This framework generated the best Pareto front, accounting for uncertainties in connection elements, and outperformed three deterministic models using nonlinear time history seismic analysis. However, it relied on a two-dimensional numerical model, limiting full three-dimensional representation, and its effectiveness depended on training data quality and diversity. Complementary research [17] applied machine learning (ML) algorithms to predict design suitability of box-shaped timber members, achieving accuracies from 91.7% to 98.6%, with Logistic Regression performing best. Both studies were limited by small datasets and constrained structural dimensions, potentially overlooking full structural variability.

### 2.2. Hybrid and CLT timber systems

The research [18] evaluated the mechanical performance of a hybrid steel-grout connection for CLT panels, using quasi-static cycle tests to assess secant stiffness and residual slip. ML models predicted these properties based on material and

geometric characteristics, highlighting rod and grout diameters, though results are specific to the tested connection and large-scale CLT assemblies. Another study [19] employed DL with a deep fully convolutional neural network (d-FCNN) using encoder-decoder architecture to segment cracks in 501 images of Yingxian Pagoda timber components. The model, trained with a batch size of six over 100 epochs, achieved average accuracy but is limited by dataset size and generalization. Anomaly detection in civil structures [20] combined Transfer Learning (TL) with Extended Node Strength Network (ENSN) to identify Regions-of-Uninterest, validated via laboratory shaking table tests. Hybrid RC-timber systems [21] for high-rise seismic zones showed up to 38.7% seismic force and 30.6% base shear reduction, though limited to numerical analysis.

## 2.3. Multi-hazard and performance-based design approaches

The study [22] evaluated the effectiveness of hybrid walls and Impact-Resilient Double Concave Frictional Pendulum (IR-DCFP) bearings in enhancing seismic performance of Light-Frame Timber Buildings (LFTBs) under severe ground motions. Using incremental dynamic analyses, archetype Chilean LFTB models were assessed for collapse margin ratios and fragility curves. The combined hybrid walls and IR-DCFP solution minimized collapse probability at Maximum Considered Earthquake (MCE) levels and provided functional isolation despite low wall density and compact bearings. Multi-hazard design frameworks [23] were tested on steel-timber hybrid connections in 18- and 36-story case buildings using Direct Displacement-Based Design (DDBD), controlling seismic drift and wind-induced accelerations, though limited to analytical and numerical modeling. Another study [24] quantified cap beams' effects on East Asian wood frames supported by stone bases, showing 42% reduction in energy dissipation, 81% increase in lateral load capacity, and 77% improvement in elastic stiffness, though applicability was limited to specific joint types and stress conditions. Table 1 provides a comparative review of notable timber-related seismic studies, emphasizing their aims, modeling methodologies, main discoveries, and current research constraints to identify knowledge gaps for future development.

## 2.4. Knowledge gap

This section exposes limits in current research on the seismic performance of wood structures, as well as a lack of integrated AI-based optimization frameworks. Table 2 provides a comparative review of important research gaps discovered in prior research on the seismic performance of wood structures, as well as a description of how the current work contributes to resolving them.

This systematic review highlights the evolution of timber seismic research, identifies limitations, and positions the GBRF-SCO framework as a robust AI-based solution addressing these critical gaps.

**Table 1. Overview of innovative research on seismic behavior of timber buildings.**

| Ref. | Objective | Model | Key Results | Limitations |
|------|-----------|-------|-------------|-------------|
| [25] | Assess seismic resistance of optimized timber structures | Topology-optimized glulam braced frame | High material efficiency; brittle response due to low redundancy | Single configuration; no experiments |
| [26] | Improve seismic stability of heritage timber | FEM with buckling-restrained braces | >50% tilt reduction; improved stiffness | Single simulation; no validation |
| [27] | Evaluate CLT seismic performance | Base-isolated CLT with FPS | High ductility; optimal seismic response | Simplified numerical model |
| [28] | Joint energy–seismic optimization | Parametric simulation framework | Strong energy–seismic interaction | Limited sites; no real-world analysis |
| [29] | Enhance wood feature detection | DL-based line segment detection | Improved accuracy and robustness | Limited generalization |
| [30] | Classify seismic damage | ML-based damage classification | High prediction accuracy | Data- and region-dependent |

**Table 2. Analysis of identified research gaps and research contributions.**

| Ref. | Research Gap | Contribution |
|---|---|---|
| [15] | DL-based CLT studies lack 3D and experimental validation | AI-optimized framework validated with real-world data for accurate 3D seismic prediction |
| [16] | ML models limited by small datasets and fixed geometries | Large, diverse dataset enables robust, generalizable seismic prediction |
| [18] | Hybrid connector studies lack scalability | Unified AI model integrates multiple connection systems for seismic optimization |
| [22] | Retrofit assessments rely mainly on simulations | Real seismic data validation ensures practical, reliable design |

## 2.5. Distinction between existing AI-based approaches and the proposed GBRF-SCO framework

Previous AI-based seismic researches on timber structures mainly rely on single learning models with manually tuned or grid-searched hyperparameters, which often suffer from high computational cost and suboptimal convergence in high-dimensional design spaces. In contrast, the proposed GBRF-SCO framework integrates a hybrid Gradient Boosted Random Forest model with the Scalable Cheetah Optimizer for adaptive, population-based hyperparameter tuning. By coupling prediction and optimization within a unified learning loop, the framework enhances nonlinear modeling capability, robustness, and design-oriented seismic performance optimization.

## 3. Methodology

An optimization framework with AI help has been developed to reliably anticipate and improve seismic performance indicators of wood structures, such as inter-story drift and roof displacement, enabling engineers to design safer and more robust timber buildings under earthquake conditions. Fig 2 depicts the GBRF-SCO framework's overall workflow for improving the seismic performance of timber structures, showing the processes of data collecting, preprocessing, feature analysis, model training, and prediction.

### 3.1. Data acquisition

The Timber Seismic Performance Dataset contains detailed information on the structural and seismic characteristics of timber buildings. It considers building height, number of storys, wall thickness, material density, and damping qualities, as well as seismic inputs such as peak ground acceleration and spectrum acceleration. Table 3 represents the collection of sample data.

The Timber Seismic Performance Dataset referenced in this work is a secondary dataset publicly available through the Kaggle repository (https://www.kaggle.com/datasets/freshersstaff/timber-seismic-performance-dataset/data). This dataset has been used under the terms and conditions provided by Kaggle. The data are accessible without restriction to any researcher, and no identifiable human subjects or sensitive personal information are included in the dataset.

### 3.2. Data preparation approaches

Data preparation for the research included tasks such as normalizing the data, detecting outliers, and extracting features. By ensuring that all input features are uniformly scaled, normalization enhances the efficiency and stability of model training. Outlier detection eliminates odd or inconsistent data points, which enhances the integrity of the dataset on the whole and is even further verified by experts familiar with the domain. The t-SNE procedure is used to extract and visualize important structural patterns and relationships among variables, which gives a clearer presentation of seismic performance characteristics.

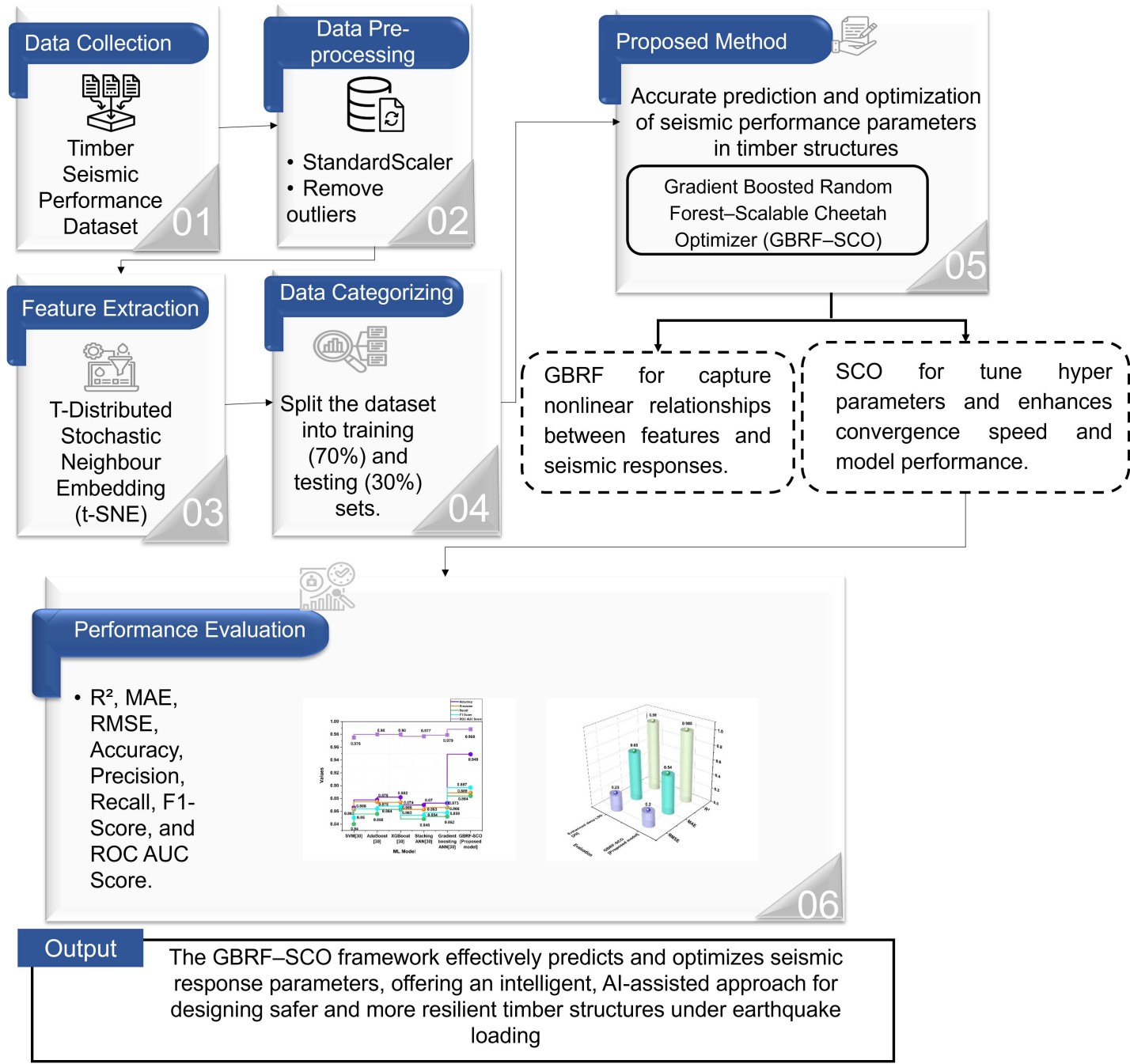

**Fig 2. The completed flowchart for enhancing seismic performance parameters of timber structures.**

**Table 3. Sample data from the timber seismic performance dataset.**

| Parameter | Sample 1 | Sample 2 | Sample 3 | Sample 4 |
|---|---|---|---|---|
| Building Height (m) | 36.21781 | 76.55 | 61.23958 | 51.90609 |
| Number of Storys | 11 | 6 | 7 | 5 |
| Story Height (m) | 3.91272 | 3.40562 | 2.6579 | 3.35902 |
| Wall Thickness (m) | 0.16088 | 0.5326 | 0.38316 | 0.45603 |
| Material Density (kg/m³) | 701.6804 | 496.9688 | 467.4328 | 557.7363 |
| Modulus of Elasticity (Pa) | 9.85E+09 | 1.14E+10 | 1.37E+10 | 9.06E+09 |
| Damping Coefficient | 0.02285 | 0.03804 | 0.01479 | 0.05187 |
| Damping Ratio per Story | 0.0332 | 0.02278 | 0.04835 | 0.03449 |
| Floor Mass (kg) | 74798.29 | 19779.63 | 77220.55 | 29065.37 |
| Natural Frequency (Hz) | 6.47606 | 7.36455 | 4.99899 | 3.67033 |
| Peak Ground Acceleration (g) | 0.24833 | 0.2076 | 0.35973 | 0.2887 |
| Spectral Acceleration (m/s²) | 2.70666 | 2.51013 | 3.1417 | 3.13824 |
| Lateral Load Resisting Ratio | 0.47871 | 0.28382 | 0.37719 | 0.78881 |
| Inter-Story Drift (m) | Timber Pegs | Hybrid Connectors | Hybrid Connectors | Timber Pegs |
| Roof Displacement (m) | Pile | Pile | Pile | Pile |
| Building Height (m) | Hip | Hip | Gable | Hip |
| Number of Storys | Shear Wall | Moment Frame | Shear Wall | Moment Frame |
| Story Height (m) | Symmetric | Asymmetric | Symmetric | Asymmetric |
| Wall Thickness (m) | 0.00246 | 0.00602 | 0.00492 | 0.00729 |
| Material Density (kg/m³) | 0.02655 | 0.03641 | 0.03649 | 0.03525 |

**3.2.1. Normalization.** The Standard Scaler assumes that information is normally distributed within each component and scales them such that the distribution is centered around 0, with a standard deviation (SD) of 1. The element's mean and SD are computed, then the component is scaled based on seismic performance parameters expressed in Equation (1).

$$Z_{scaled} = \frac{x - \mu}{\sigma}$$

(1)

Normalized data by subtracting the mean ($\mu$) and dividing by the standard deviation ($\sigma$), ensuring all features are on a comparable scale for improved model performance.

**3.2.2. Outlier detection.** The Isolation Forest algorithm is used to find anomalies in the seismic and structural dataset by looking for samples that are different from the rest. The contamination parameter is set to 0.05, which means that about 5% of the observations are likely to be outliers. Samples with anomaly scores below the decision threshold are considered anomalous and taken out of the dataset because they show physically inconsistent or severe seismic–structural pairings. This method cuts down on noise and bias, makes the data more reliable, and ensures that the samples kept are a true representation of how timber structures behave in genuine seismic conditions. This makes model training more stable and effective.

**3.2.3. Feature analysis using t-distributed stochastic neighbor embedding (t-SNE).** This method enables feature extraction through dimensionality reduction, or the compression of data while preserving the connections between data points through the seismic performance criteria. Visualizations of correlations enable better decision-making by transforming convoluted data into its patterns and illuminating similarities, correlations, and clusters between points in an informative manner. By means of conditional probabilities, t-SNE augments standard geometrical distances to represent

these similarities. The conditional probabilities $O_{i|j}$, which are stated in Equation (2), express that similar data points $w_j$ and $w_i$ became individuals.

$$O_{i|j} = \frac{\exp\left(\frac{-||w_j-w_i||^2}{2\sigma_j^2}\right)}{\sum_{l\neq j}\exp\left(\frac{-||w_j-w_i||^2}{2\sigma_j^2}\right)}$$

(2)

It calculates the conditional probability $O_{i|j}$, showing how similar point i is to point j. Here, $w_i$ and $w_j$ are their feature vectors, and $\sigma_j$ controls how distance affects similarity. The numerator measures how close the two points are, while the denominator ensures all probabilities around j sum to one. This helps t-SNE identify clusters of similar data points. In the original space, the probabilities were specified by Equation (3).

$$O_{j,i} = \frac{(O_{j|i} + O_{i|j})}{2m}$$

(3)

It defines the joint probability $O_{j,\,i}$, which represents the combined similarity between points i and j. It is calculated as $O_{j,i} = \frac{O_{j|i}+O_{i|j}}{2m}$. Here, $O_{j|i}$ and $O_{i|j}$ are the conditional probabilities measuring how similar each point is to the other, and m is the overall data points. It ensures the similarity measure is symmetric, meaning the relationship between two points is treated equally in both directions. Where 2m represents the dataset's size. The smooth indicator of an efficient number of neighbors represents the perplexity parameter that the t-SNE algorithm takes as an input, as shown in Equation (4).

$$\text{Perp}\,(O_j) = 2^{G(O_j)}$$

(4)

It defines the perplexity of point $O_j$ as $\text{Perp}(O_j) = 2^{G(O_j)}$. $G(O_j)$ represents the Shannon entropy of the probability distribution for point j. Perplexity indicates how broadly the probability distribution spreads across nearby points, effectively showing the close neighbors. A greater perplexity value means that more neighboring points are considered when measuring similarity. Here, $G\,(O_j)$ is the bit value of the Shannon entropy $O_j$, as shown in Equation (5).

$$G\,(O_j) = -\sum_i O_{i|j}\log_2 O_{i|j}$$

(5)

It defines the Shannon entropy $G(O_j)$ as $G(O_j) = -\sum_i O_{i|j}\log_2 O_{i|j}$. In this equation, $O_{i|j}$ represents the conditional probability that point i is similar to point j. The entropy $G(O_j)$ measures the uncertainty or spread of these probabilities. A higher entropy value indicates that the similarities are distributed more evenly among many points, while a lower value means that only a few points are strongly similar to j. Equation (6) defines the probabilities at low-dimensional $r_{ji}$ using this distribution.

$$r_{ji} = \frac{\left(1 + ||z_j - z_i||^2\right)^{-1}}{\sum_{l\neq k}\left(1 + ||z_l - z_k||^2\right)^{-1}}$$

(6)

$z_j$ and $z_i$ are the low-dimensional representations of data points j and i. The numerator, $(1+ |z_j - z_i|^2)^{-1}$, measures how close the two points have a higher similarity. The denominator $||z_l - z_k||$ ensures that all probabilities in the low-dimensional integral sum to one. The t-SNE technique determines the lower-dimensional projection of the input data $z_j$ as $z_i$, therefore reducing the divergence among $o_{ji}$ and $r_{ji}$. After data preparation, the data is divided into two categories: testing (30%) and training (70%).

 

### 3.3. Gradient boosted random forest machine with scalable cheetah optimizer (GBRF-SCO) for accurate seismic predictions in timber structures

The GBRF-SCO framework employs an AI-based predictive ensemble combining gradient boosting and random forest to model nonlinear seismic–structural relationships. SCO adaptively optimizes GBRF hyperparameters using a cheetah-inspired search strategy, enhancing prediction accuracy, stability, convergence, and computational efficiency, thereby supporting the design of safer, more earthquake-resilient timber structures.

**3.3.1. Gradient boosted random forest (GBRF) for accurate seismic response prediction.** The GBRF model merges aspects of gradient boosting and random forest methods. The model successfully captures the intricate, nonlinear connections between seismic and structural events. The linear combination of many weak learners increases overall prediction accuracy and generalization. This model is used for estimating inter-story drift and roof displacement in timber structures.

#### • Gradient Boosting (GB) algorithm

The GB improves prediction accuracy by successively merging many weak learners to reduce residual errors, making it particularly efficient at capturing complicated nonlinear correlations between structural and seismic data. The GB architecture is represented in Fig 3.

The GB algorithm can approximate the underlying function $E(w)$ given an input matrix $w$ and a vector of molecular properties. This function maps the relationship between the molecular descriptor and the biological activity. The function $\hat{E}(w)$ is constructed in an additive manner. Equation (7) is used in gradient boosting to combine multiple weak learners into a strong predictive model by gradually minimizing the overall error.

$$\hat{E}(w) = \sum_{n=1}^{N} \sigma * \hat{E}_n(w)$$

(7)

Here, $N$ is the total number of iterations, $\hat{E}(w)$ represents the total error of the model, $\hat{E}_n(w)$ is the error from the $n^{th}$ iteration or tree, and $\sigma$ is the rate of learning that determines how much each new model contributes to the last prediction. After the first iteration, minimize the following goal given a loss function, which assesses the quality of predictions pi in relation to real readouts. Equation (8) defines the optimization step for the $n^{th}$ iteration in the gradient boosting process.

$$\hat{E}_n = \text{argminF}\left(\frac{-\partial K(Z, O_{n-1})}{\partial O_{n-1}} - O_n\right)$$

(8)

$\hat{E}_n$ represents the minimized error function at iteration n, and $K(Z, O_{n-1})$ denotes the loss function that calculates the variations among the predicted values $O_{n-1}$ and the true outcomes $Z$. The term $\frac{\partial K(Z, O_{n-1})}{\partial O_{n-1}}$ is the gradient of the loss function with respect to the previous prediction, indicating the direction of steepest increase in error. By taking the negative gradient, the model updates in the direction that reduces the error most effectively. The $\text{argmin}_F$ expression means that the algorithm searches for the function F that minimizes this difference. In simple terms, this equation guides the model to adjust its predictions step by step to reduce the overall error during training.

#### • Random Forest (RF) classifier

The RF enhances model resilience and generalization by combining findings from several Decision Trees (DTs), decreasing overfitting, and ensuring consistent predictions of seismic reactions in various timber structure configurations. The recommended binary classification strategy makes use of the RF algorithm. Following the training phase, RF builds a huge number of decision trees and produces a class with an average prediction. RF hyperparameters were changed using grid search. Fig 4 denotes the architecture of the RF classifier.

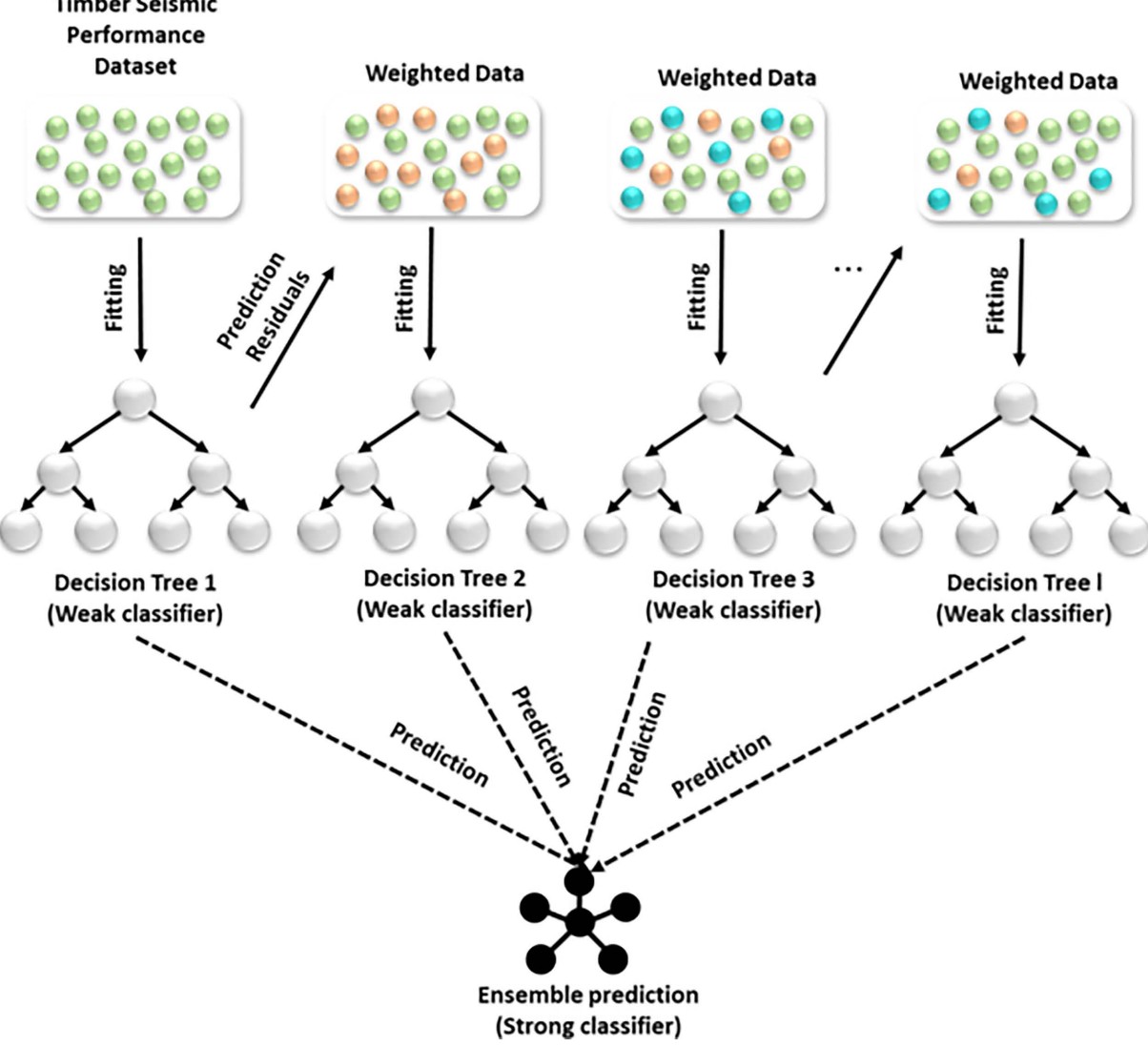

**Fig 3. The schematic representation of GB decision tree.**

This research uses 1000 random trees with a maximum depth of 10, utilizing a 0.5 confidence vote method and the Gini impurity criterion. The Gini impurity can be determined as follows in equation (9):

$$H = \sum_{i=1}^{D} O(i) * (1 - o(i))$$
(9)

Here, $O(i)$ represents the data point probability that belongs to class i, and D is the overall classes. The term $O(i)\,(1 - O(i))$ measures how mixed or impure the data is for each class. When all data points belong to one class, H becomes zero, indicating perfect purity. Higher values of Hmean indicate greater uncertainty or diversity among the classes.

### 3.3.2. Traditional Cheetah Optimizer (TCO).
The TCO is a cheetah-inspired metaheuristic optimizing timber structure parameters to enhance seismic performance by balancing global exploration and local exploitation. It employs

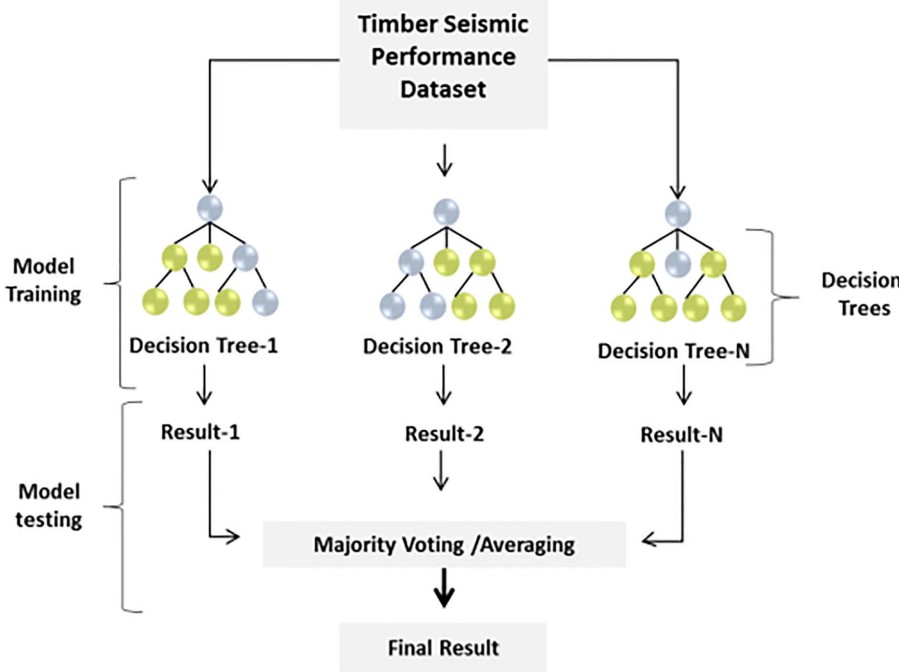

**Fig 4. The general architecture of the RF classifier.**

three strategies to avoid local optima. Fig 5 illustrates SCO phases: initial exploration, transition, attacking, and final convergence, systematically guiding the search toward optimal solutions. In the figure, $W_{j,i}^{S0}$ represents the initial position of the cheetah in the search space, and $W_{j,i}^{S1}$, $W_{j,i}^{S2}$, $W_{j,i}^{S3}$, and $W_{j,i}^{S4}$ are its updated positions during different hunting phases. $q_{j,i}$ and $\widetilde{q}_{j,i}$ are random factors controlling movement, while $\alpha_{j,i}^{S1}$ and $\beta_{j,i}^{S2}$ are adaptive coefficients that balance exploration and convergence. $W_{A,i}^{S2}$ denotes the best or attacking position guiding the cheetah toward the optimal solution.

**Scalable Cheetah Optimizer (SCO) for Timber Structure Seismic Parameter Optimization:** The Scalable Cheetah Optimizer (SCO) is a metaheuristic optimization algorithm designed for efficient hyperparameter tuning and seismic parameter optimization in timber structures. The SCO extends the traditional Cheetah Optimization (CO) framework by introducing adaptive control mechanisms, scalable step-size updates, and opposite learning concepts, ensuring high accuracy, rapid convergence, and improved stability in high-dimensional optimization problems.

**Initialization of individuals:** The optimization process begins with initializing the positions of cheetahs (candidate solutions) in a d-dimensional search space. The initial value of the $i^{\text{th}}$ variable for the j-th individual, $W_{j,i}$, is defined as Equation (10)).

$$W_{j,i} = \text{LB}_i + q \cdot (\text{UB}_i - \text{LB}_i), j = 1, 2, \ldots, r; \ i = 1, 2, \ldots, c \tag{10}$$

where $\text{LB}_i$ and $\text{UB}_i$ are the lower and upper bounds of the i-th variable, r denotes a random number, c is the total number of variables, and q is a uniform random number between 0 and 1. This ensures a uniform random distribution of initial solutions across the defined search space.

**Searching technique:** During the search phase, cheetahs evaluate their environment, including prey location, cover, and their own status. Each individual's position is updated according to the same initialization principle (Equation 11), maintaining randomness in early exploration. The step size $\alpha_{j,i}^s$ is dynamically adapted at each iteration based on the current iteration ratio $\frac{s}{S}$, the distance between candidate solutions, and random perturbations expressed in Equation (11).

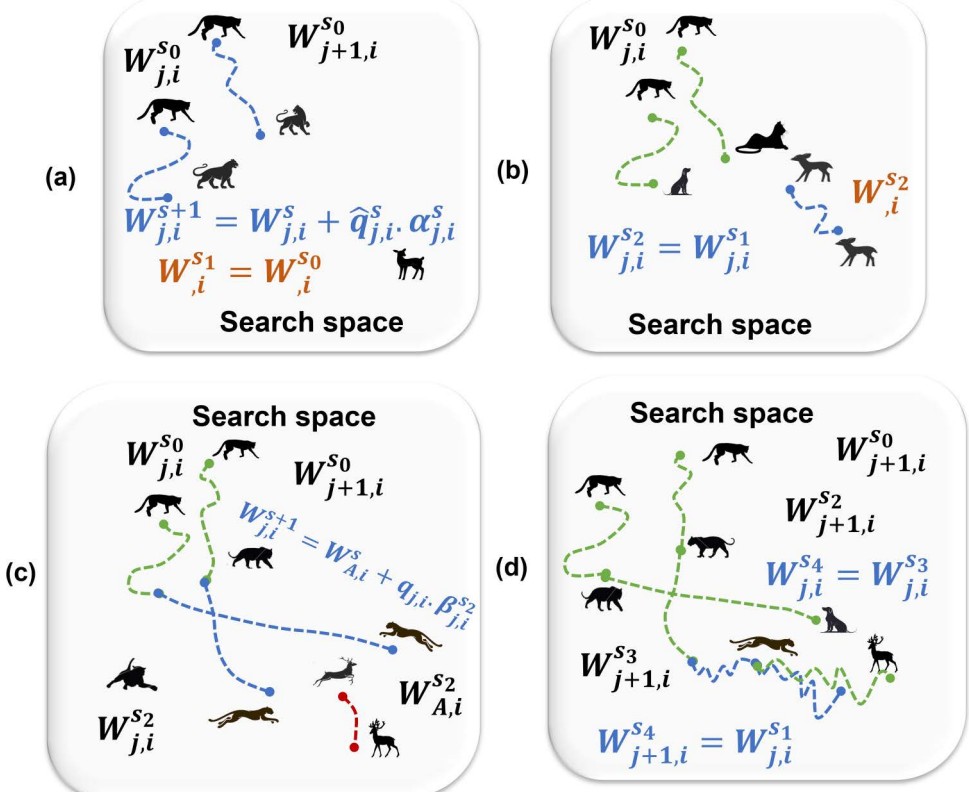

**Fig 5. The optimization process of SCO (a) the exploration phase, where cheetahs search for potential solutions, (b) transition phase moving toward promising regions, (c) the attacking phase with rapid convergence to the best solutions, and (d) depicts the final convergence phase, where the algorithm stabilizes around the optimal result.**

$$\alpha_{j,i}^{s} = \begin{cases} 0.001\frac{s}{S}(UB - LB) & I = 1 \\ 0.001\frac{s}{S}\left|W_t + \frac{b_1}{b_3+b_2} \times \left[W_a^s(j,i) - \overline{W}_j^I\right] + \frac{b_2}{b_1+b_2} \times \left[W^s(j,i) - \overline{W}_j^I\right] - W^s(j,i)\right| + q \text{ and} & I \neq 1 \end{cases} \tag{11}$$

It describes the adaptive control parameter $\alpha_{(j,\ i)}^s$, which adjusts the step size for each variable during the optimization process. When $I = 1$, the value of $\alpha_{(j,\ i)}^s$ is determined by the ratio of the current iteration. The overall number of iterations S, multiplied by the variable range $(UB - LB)$. When $I \neq 1$, the value depends on the distance between the target position $W_t$, current position $W^s(j,\ i)$, and other reference positions like $W_a^s(j,i)$ and $\dot{W}_j^I$. The constants $b_1$, $b_2$, and $b_3$ control the weight of each term, while q adds a variation's random factor.

**Sitting-and-Waiting Model:** The SCO also incorporates a sit-and-wait strategy, reflecting the cheetah's behavior of remaining stationary to reduce unnecessary movements are expressed in Equation (12).

$$W_{j,i}^{s+1} = W_{j,i}^{s} \tag{12}$$

Here, the variable remains unchanged between iterations, allowing the algorithm to conserve computational resources while maintaining candidate diversity.

**Attacking Strategy:** During the attacking phase, cheetahs adjust positions dynamically based on prey, nearby individuals, and the leader's position. The position update is expressed as Equation (13 and 14).

$$W_{j,i}^{s+1} = W_{A,i}^{s} + \bar{q}_{j,i} \cdot \beta_{j,i}^{s} \tag{13}$$

$$\beta_{j,i}^{s} = W_1^{s}(j,i) - W^{s}(j,i) \tag{14}$$

where $W_{A,i}^{s}$ is the leader or reference position, $\bar{q}_{j,i}$ is a random scaling factor, and $\beta_{j,i}^{s}$ represents the difference between the best solution and the current candidate. This allows efficient exploitation of promising areas in the search space.

**SCO for Hyperparameter Optimization:** The SCO framework ensures fast and reliable convergence in predicting seismic parameters by dynamically adjusting GBRF hyperparameters. Unlike the traditional TCO algorithm, which suffers from slow convergence, early stagnation, and inefficiency in high-dimensional problems, SCO is adaptive, scalable, and provides improved optimization performance. The opposite learning strategy further enhances exploration in Equation (15).

$$\widetilde{W}_j^l = \gamma \cdot (k + v) - W_j^l \tag{15}$$

where $\widetilde{W}_j^l$ is the opposite position of the j-th individual in dimension l, $\gamma$ is a scaling factor, and k, v defines the exploration center.

**Control Parameter Update Strategy:** The CO algorithm updates control parameters adaptively to match nonlinear optimization requirements. The step size $\alpha_{j,i}^{s}$ is updated as Equation (16 and 17).

$$\alpha_{j,i}^{s} = 0.001 \cdot \frac{s}{S}(\text{UB} - \text{LB}) \text{for } l = 1 \tag{16}$$

$$\beta_{j,i}^{s} = W_t + \frac{b_3}{b_3 + b_2} \cdot (W_1^{s}(j,i) - \bar{W}_j^l) + \frac{b_2}{b_3 + b_2} \cdot (W^{s}(j,i) - \bar{W}_j^l), l \neq 1 \tag{17}$$

Here, $W_t$ is the target position, $W_1^{s}(j,i)$ the best-known solution, and $\bar{W}_j^l$ the reference position. Constants b2 and b3 control the contribution of each term, while q introduces randomness. This strategy balances global exploration with local fine-tuning.

**Updating Individual Locations:** To enhance individual updates and prevent redundant searches, the SCO employs the following update mechanism for binary and probabilistic optimization is expressed in Equation (18).

$$W_{j,i}^{s+1} = \frac{1}{1 + \exp(-\bar{W}_{j,i}^{s+1})} \tag{18}$$

The sigmoid function bounds updated values between 0 and 1, suitable for probabilistic decisions. For discrete optimization is compute as Equation (19).

$$W_{j,i}^{s+1} = \begin{cases} -1, & q > -\bar{W}_{j,i}^{s+1} \\ 1, & \text{otherwise} \end{cases} \tag{19}$$

This converts continuous positions to binary states, enabling SCO to handle both continuous and discrete seismic optimization problems effectively. In simple terms, this decides whether each variable can take the value –1 or 1 in the next iteration. The SCO algorithm employs initialization, searching, and attacking strategies with adaptive control, enhancing convergence, stability, and scalability while optimizing GBRF hyperparameters, improving accuracy and efficiency in seismic parameter optimization for timber structures. Table 4 defines the key hyperparameters and their mathematical formulations used in the GBRF-SCO for seismic performance prediction.

 

**Table 4. Hyperparameter for GBRF and SCO optimization framework.**

| Category | Hyperparameters | Variables | Values |
|---|---|---|---|
| GBRF | Number of Trees | $N$ | 50–300 |
| | Learning Rate | $\sigma$ | 0.01–0.3 |
| | Maximum Depth | $d$ | 2–8 |
| | Loss Function | $K(Z, O)$ | Mean Squared Error (MSE) |
| | Error Function | $\hat{E}(w)$ | Computed value |
| SCO | Population Size | $r$ | 8–10 |
| | Maximum Iterations | $S$ | 15–20 |
| | Step Size | $\alpha$ | (0.001 ×(s/ S)) |
| | Position Difference | $\beta$ | Dynamic |
| | Opposite Position Factor | $\gamma$ | 0.5 |
| | Lower Bound | $LB$ | Defined per variable |
| | Upper Bound | $UB$ | Defined per variable |

The selection of SCO hyperparameters depends on how well they converge, how fast they can be computed, and how many dimensions the seismic optimization problem has. The population size (r = 8–10) makes sure that there are enough different solutions without making the computation too expensive in feature spaces with a lot of dimensions. Smaller populations limit research, but bigger ones provide slight advances in accuracy. Convergence analysis, which is when the optimization error stabilizes, tells you the maximum number of iterations (S = 15–20). This setup strikes a good mix between exploration and exploitation, which makes GBRF hyperparameter tuning steady and dependable.

```
Pseudo code 1 represents the GBRF-SCO framework, which integrates gradient boosting for learning
seismic behaviour with the SCO to adaptively tune model parameters, achieving accurate and efficient
seismic performance prediction for timber structures.
Pseudo Code 1: GBRF–SCO Framework for Seismic Performance Prediction in Timber Structures
Input:
X = [x₁, x₂, ..., xₘ]          # Structural and seismic features
Y = [y₁, y₂, ..., yₘ]          # Continuous seismic response (roof displacement / drift)
P                              # SCO population size
S                              # SCO iterations
LB, UB                         # Hyperparameter bounds
Output:
Ŷfinal                         # Optimized seismic response prediction
Step 1: Data Preprocessing
    1. Remove outliers using Isolation Forest
    2. Normalize features using Robust Scaling
    3. Split data into training and testing sets
Step 2: Gradient Boosted Random Forest (GBRF) Modeling
    1. Initialize GBRF hyperparameters:
θ = {nestimators, η, dmax}
    2. Train GBRF model on (Xtrain, Ytrain)
    3. Predict seismic response Ŷ
Step 3: Scalable Cheetah Optimizer (SCO)
    1. Initialize the cheetah population:
Wⱼ⁰ = LB + rand(0, 1) × (UB − LB)
    2. For s = 1 to S:
Evaluate fitness:
Fitness(Wⱼˢ) = RMSE(Y, Ŷ)
Identify the best cheetah W*
```

*Update position:*

$$W_j^{s+1} = \sigma \left( \gamma(k + v) - W_j^s + \alpha_s(W^* - W_j^s) \right)$$

*Apply boundary constraints*
 *3. Return optimal hyperparameters $\theta^*$*
**Step 4: Final Prediction**
 *1. Retrain GBRF using $\theta^*$*
 *2. Generate optimized seismic response: $\hat{Y}_{final}$*
**Step 5: Seismic Damage Classification (Optional)**

$$Class = \begin{cases} Low, & \hat{Y}_{final} < Q_{33} \\ Medium, & Q_{33} \leq \hat{Y}_{final} < Q_{66} \\ High, & \hat{Y}_{final} \geq Q_{66} \end{cases}$$

**Return:** $\hat{Y}_{final}$

The GBRF-SCO framework partitions data, learns structural–seismic relationships by minimizing errors, and optimizes hyperparameters via cheetah-inspired SCO. Repeated learning identifies optimal parameters, while MLR assesses key factors, enabling the retrained GBRF to predict accurate, reliable seismic performance in timber structures.

## 4. Performance analysis and discussion

The objective is to provide an AI-assisted optimization framework for precisely predicting and improving the seismic performance characteristics of wood buildings, with an emphasis on inter-story drift and roof displacement. Table 5 presents the hardware and software specifications used for implementing and testing the GBRF-SCO framework.

### 4.1. Multiple linear regression analysis for evaluating seismic performance relationships

The MLR approach analyzes the effects associated with different structural and seismic indicators, such as building height, wall thickness, and the acceleration of motion on the roof displacement of timber buildings. Table 6 presents the MLR results showing how key structural and seismic parameters influence roof displacement and GBRF-SCO's overall seismic performance in timber structures.

### 4.2. Distribution of roof displacement by structural system type

Determine whether there are structural system-specific differences in the movement of roofs subjected to seismic forces. In order to determine which system can withstand an earthquake better in terms of stability and resilience. Better

**Table 5. Experimental setup for GBRF-SCO model implementation.**

| Category | Specification | Details |
|---|---|---|
| Hardware Configuration | Processor | Intel Core i7, 3.4 GHz |
| | RAM | 16 GB DDR4 |
| | ROM/ Storage | 512 GB SSD |
| | System Type | 64-bit Operating System |
| Software Configuration | Operating System | Windows 11/ Ubuntu 22.04 |
| | Programming Language | Python 3.10 |
| | Development Environment | Jupyter Notebook/ VS Code |
| | Libraries Used | NumPy, Scikit-learn, Matplotlib, Pandas |
| | Optimization Framework | SCO implemented in Python |

**Table 6. MLR analysis of seismic parameters.**

| Independent Variable | T Statistic | Standard Error | p-Value | Significance |
|---|---|---|---|---|
| Intercept | 3.95 | 0.0021 | 0.0001 | *** |
| Building Height | 5.25 | 0.00004 | 0.0000 | *** |
| Number of Storys | 4.22 | 0.00027 | 0.0000 | *** |
| Wall Thickness | −2.94 | 0.00182 | 0.0035 | ** |
| Material Density | −2.00 | 0.000002 | 0.0470 | * |
| PGA | 4.30 | 0.0102 | 0.0000 | *** |
| SA | 2.81 | 0.0031 | 0.0052 | ** |
| Lateral Load Resisting Ratio | −2.16 | 0.0058 | 0.0320 | * |
| Damping Coefficient | −2.77 | 0.0315 | 0.0063 | ** |
| Model Summary | | | | |
| $R^2 = 0.921$ | $p < 0.001$ | F (8, 120) = 56.82 | Durbin–Watson = 1.95 | — |

**Note:** The MLR model achieved excellent fit ($R^2 = 0.921$), explaining 92.1% of roof displacement variation. Roof displacement increased with Building Height and PGA, while greater Wall Thickness and Damping Coefficient reduced it, with significance levels indicated by ($p < 0.001$), ** - moderately significant predictors ($p < 0.01$), * - significant predictors ($p < 0.05$),

knowledge of structural behavior and more efficient design of wood structural systems lead to stronger buildings. Fig 6 shows the distribution of roof displacement among structural system types under seismic loading.

These findings suggest that the GBRF-SCO model may provide a good representation of the structural system-specific variation in roof displacement. By consistently outperforming the others in displacement, the hybrid system proves that the suggested framework improves the precision of seismic performance forecasts.

### 4.3. Comparative analysis of seismic ground and structural parameters

The research examines the consequences that various seismic intensity parameters have on the lateral load resistant capabilities of varying buildings, focusing on the influence these parameters have on the stability, strength, and seismic performance of residential structures, which would assist in optimizing structural performance for diverse ground motions, allowing more factual assessment of overall earthquake resistance, depicted in Fig 7 for each structural sample under seismic acceleration characteristics and lateral load resisting ratios.

The patterns that overlap indicate a high correlation between the intensity of ground motion and the load-resisting capability of the structure. There is a little positive correlation between building height and roof displacement, as shown in Fig 8(a) by the trend line. Fig 8(b) Roof displacement versus peak ground acceleration is shown, with materials color-coded according to density. Fig 8(c) shows the overall patterns of fluctuation in roof displacement as a function of data indices, including the rolling mean variation.

The findings suggest that there is a connection between essential seismic parameters and roof displacement. The maximum ground acceleration varies from lightly felt to moderately felt to strongly felt; this is also true for roof displacement, which experiences a slight increase with greater height. The overall oscillation trends of the series of timber building samples reinforce some general consistent structural performance and stability of the GBRF-SCO.

### 4.4. Comparative analysis of roof displacement and structural feature relationships for different roof types

To investigate how roof geometry and critical structural characteristics affect roof displacement and overall seismic behavior, the study analyzes their influence on structural stability, deformation patterns, and the seismic performance of timber buildings. Fig 9(a) shows the variation of roof displacement with the number of storys for different roof types, indicating

 

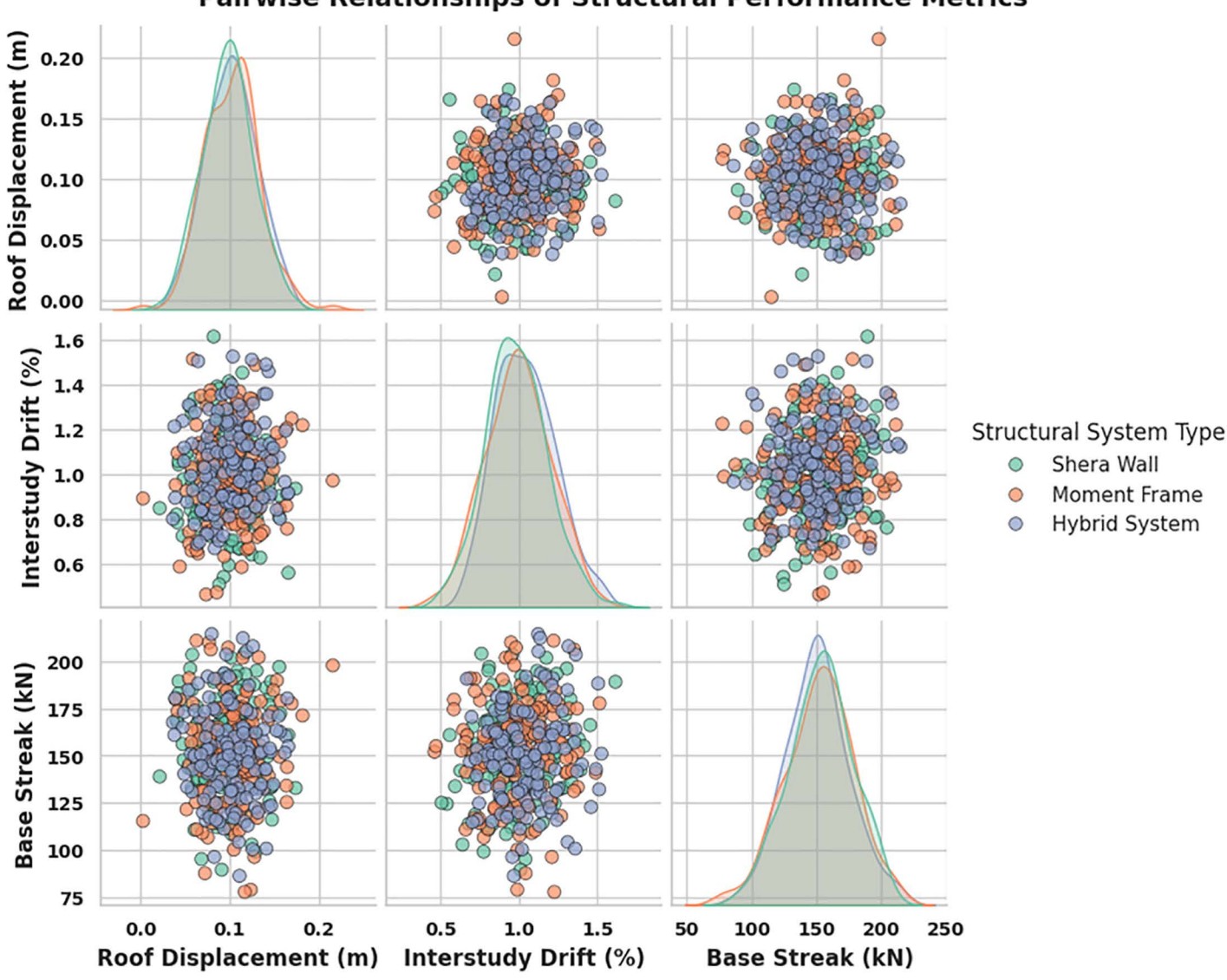

**Fig 6. Roof displacement variation for timber structural system types.**

that displacement generally increases with height. Fig 9(b) illustrating relationships among structural features and roof displacement.

The results show that the geometry of the roof has a major impact on the earthquake reaction, especially for sawtooth and hip roof types, and that more storys result in more displacement of the roof. Roof displacement is also significantly affected by wall thickness and material density, according to the study, which shows how geometry and material parameters work together to affect seismic performance.

### 4.5. Pairwise correlation analysis of key structural and seismic parameters

The examination is to explore interrelationships among structural parameters such as height, storys, wall thickness, and material density, and their combined influence on roof displacement under seismic loading. Fig 10 showing relationships

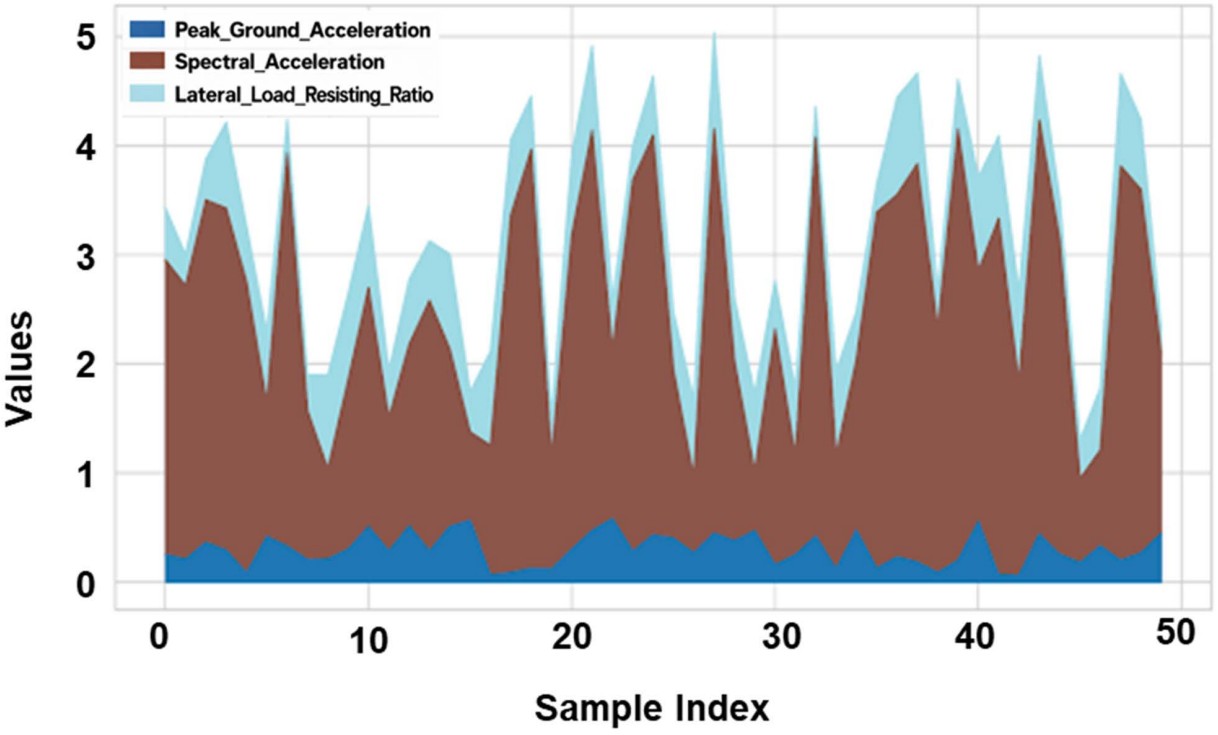

**Fig 7. Stacked Area Plot of Seismic Parameters.**

between building and material parameters with roof displacement, helping identify key factors influencing seismic performance.

The findings indicate that roof displacement rises modestly with building height and number of storys, although wall thickness and material density have less association in GBRF-SCO. This suggests that geometric characteristics have a stronger effect on seismic reaction than material attributes in wood constructions.

### 4.6. Assessment of interdependencies among seismic response variables

Roof displacement and inter-story drift are two seismic and structural variables that the GBRF-SCO hopes to shed light on. These variables include building height, material properties, and ground acceleration. The correlation heatmap shown in Fig 11 illustrates the strength and direction of connections among structural, material, and seismic elements.

There is a considerable positive connection between building height (0.35–0.54) and the number of storys (0.66), and the results show that there is a strong association (0.67) between inter-story drift and roof displacement.

### 4.7. Assessment of inter-story drift and roof displacement consistency trends

To assess the patterns of structural deformation, compare the inter-story drift with the movement of the roof. This shows that the structure is generally stable and resilient, which is consistent with the GBRF-SCO's evaluation of the seismic forces' propagation through wood structures. See how inter-story drift compares to roof deformation in wood buildings in Fig 12.

The result illustrates that the values of roof displacement are consistently greater than those of inter-story drift, suggesting that lateral deformation accumulates along the building's height under seismic loading conditions.

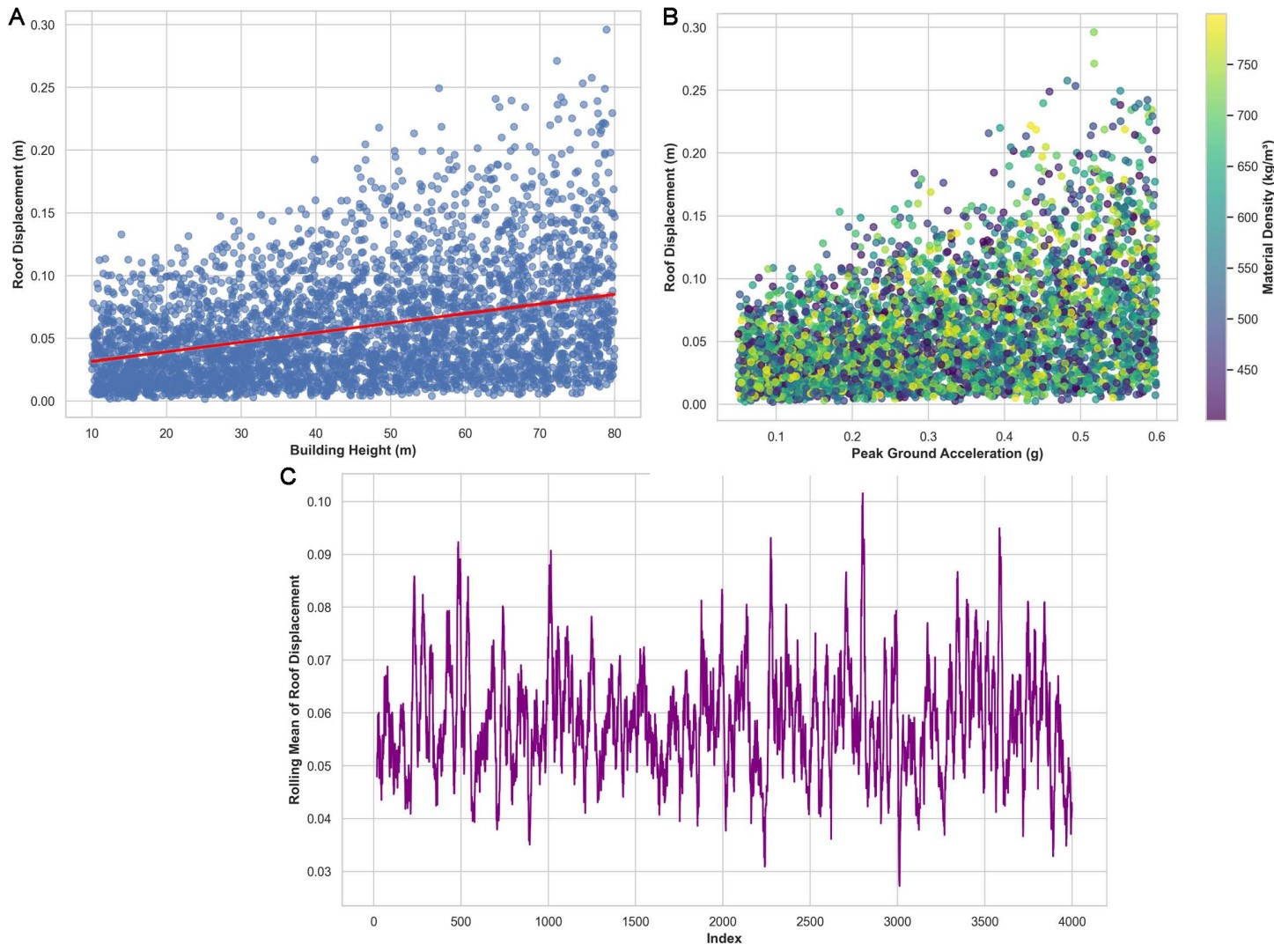

**Fig 8. Findings of (a) Relationship between building height and roof displacement, (b) Variation of roof displacement with peak ground acceleration, (c) Trend of roof displacement across samples showing overall fluctuations patterns.**

#### 4.8. Classification performance of roof displacement categories using a confusion matrix

The investigation verifies that the model accurately predicts the degree of roof displacement, which is necessary for correctly categorizing structural seismic reactions. Fig 13 displays the results of the roof displacement classes' classification performance.

Based on quantile thresholds from the dataset's empirical distribution, roof displacement values are divided into three ordinal classifications such as low, medium, and high. Values of displacement that are less than the 33rd percentile are considered low, values that are between the 33rd and 66th percentiles are considered medium, and values that are greater than the 66th percentile are considered high. This classification, based on quantiles, keeps the class proportions even, keeps the ordinal structure of seismic reaction severity, and doesn't let people choose the threshold. The established criteria provide a uniform interpretation of the confusion matrix and elucidate the lack of significant misclassifications between low and high displacement categories.

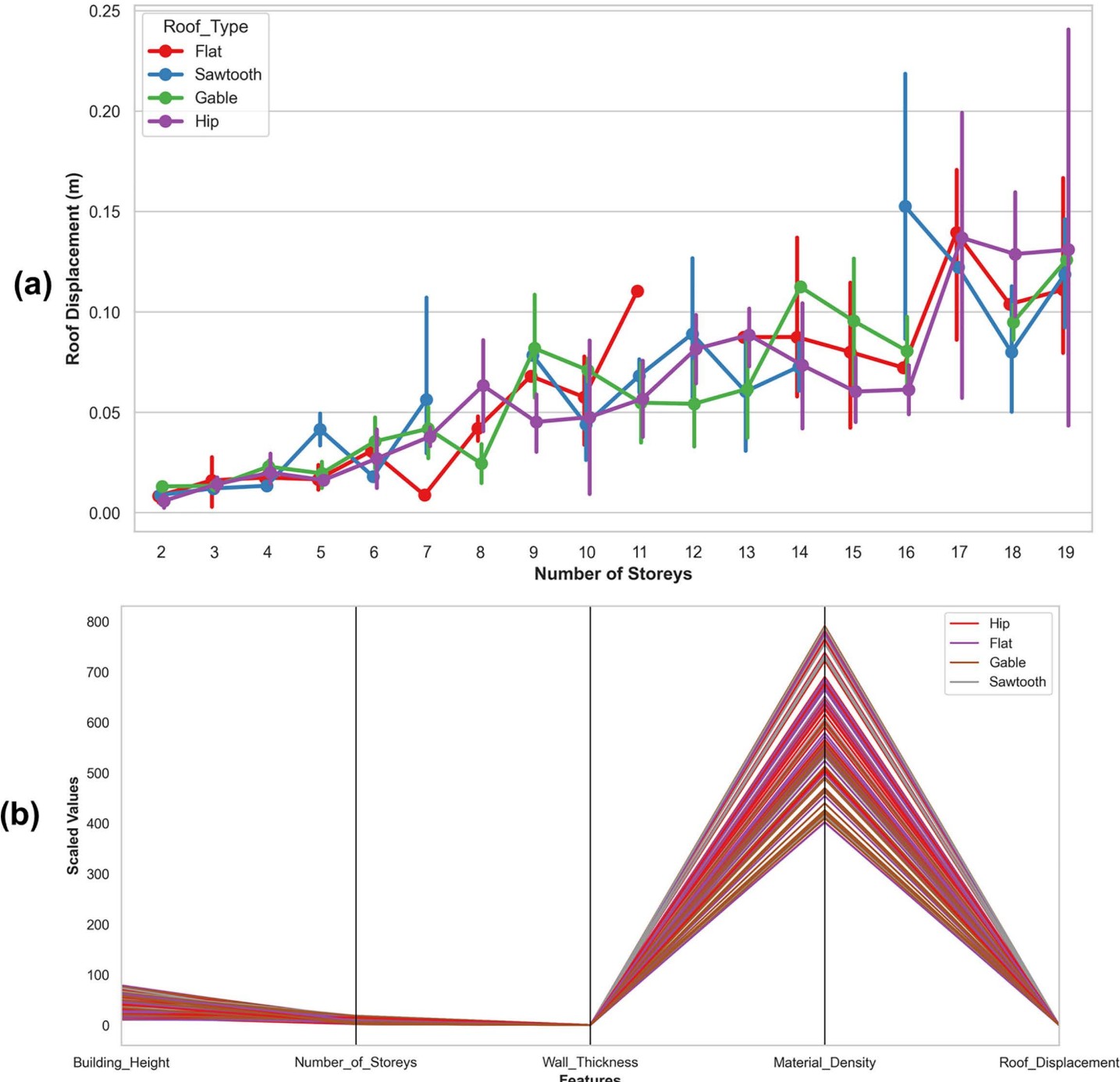

**Fig 9. Graphical findings of (a) roof displacement with number of storys for different roof types, and (b) Parallel coordinate analysis of structural features influencing roof displacement across rooftypes.**

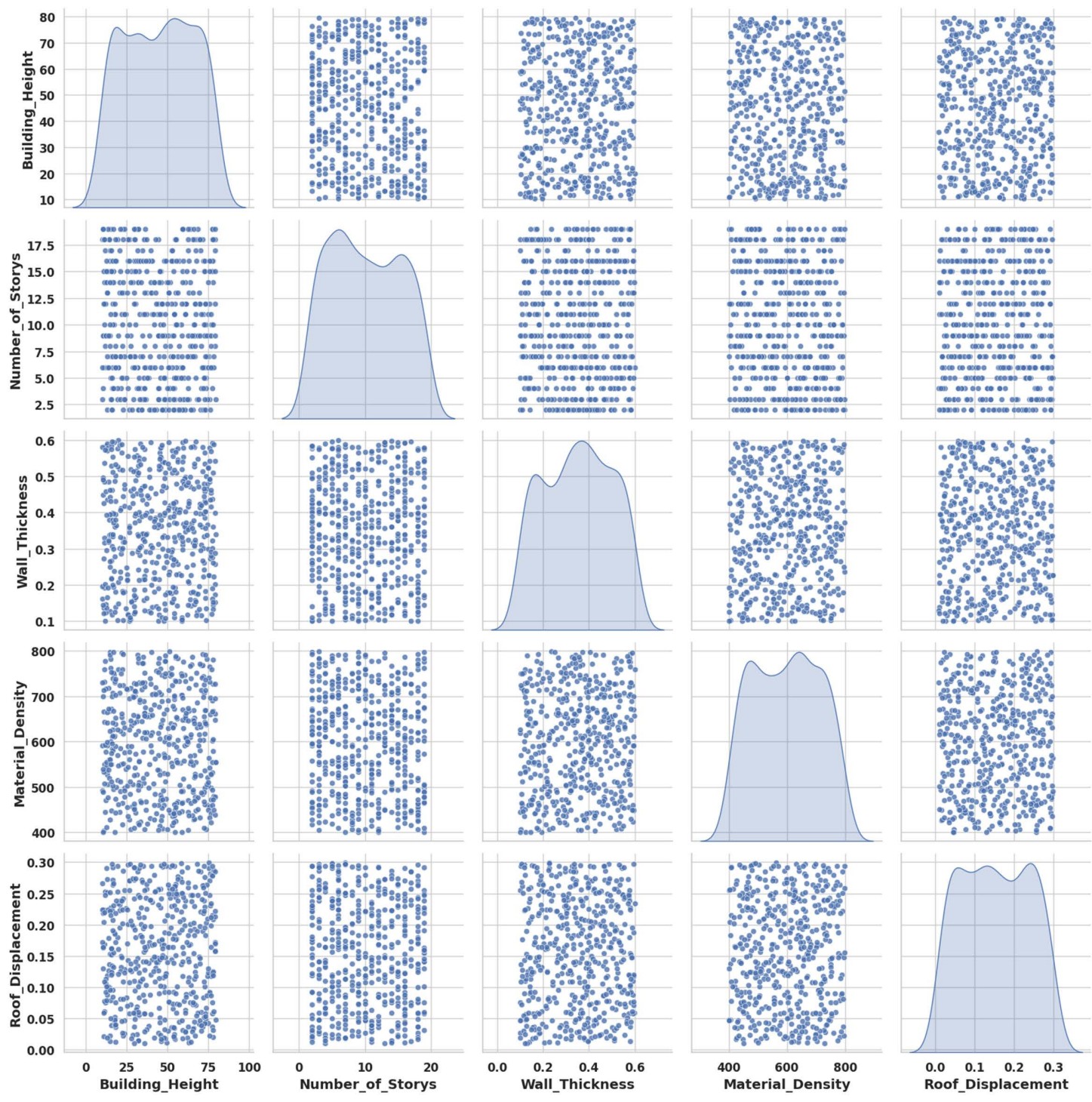

**Fig 10. Pairwise correlation analysis of key structural and seismic parameters.**

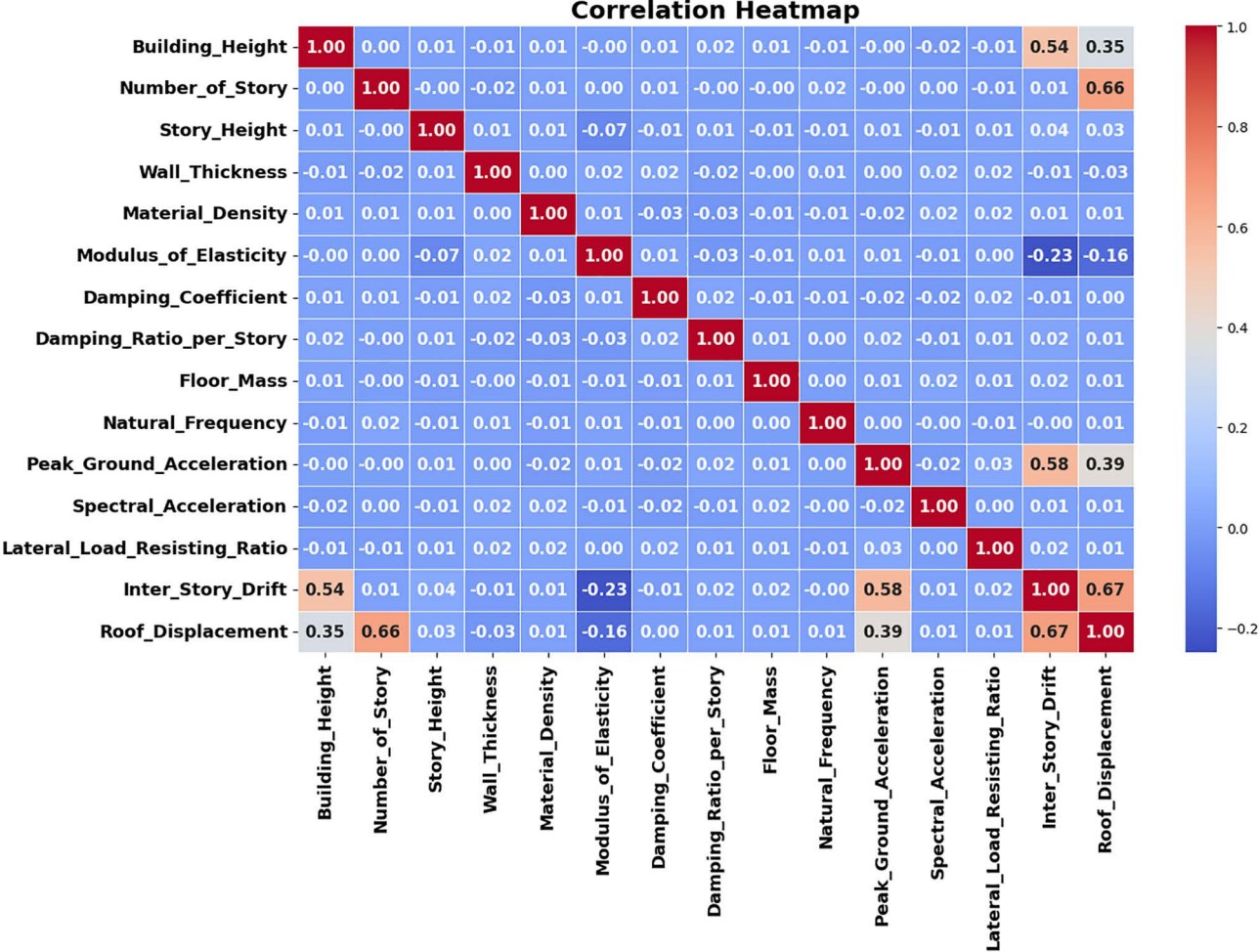

**Fig 11. Correlation analysis of seismic and material parameters in timber structures.**

A small number of incorrect classifications show that the model can distinguish between wood structures with varying degrees of seismic reactivity when using GBRF-SCO. During seismic parameter prediction, Fig 14 shows that the model learned steadily and converged strongly.

According to the results, the suggested GBRF-SCO model steadily increased testing and training accuracy across all epochs. To make the accuracy and loss trends easier to see, the legend was moved outside the graph. The GBRF-SCO model achieved a final training accuracy of 94.9% by displaying strong convergence and little validation variance.

## 4.9. Comparative analysis

This experiment evaluates the accuracy and efficiency of prediction models for seismic performance in timber structures, comparing the proposed Gradient Boosting Random Forest with Scalable Cheetah Optimizer (GBRF-SCO) to existing methods, including Enhanced Deep Line Segment Detection (LSD) [29], Support Vector Machine (SVM) [30], Adaptive Boosting (AdaBoost) [30], Extreme Gradient Boost (XGBoost) [30], Stacking Artificial Neural Network (ANN) [30], and Gradient Boosting ANN [30]. Performance is assessed using R², Mean Absolute Error (MAE), Root Mean Squared Error (RMSE), Accuracy, ROC-AUC, Precision, Recall, and F1-Score, ensuring a comprehensive evaluation of predictive fit,

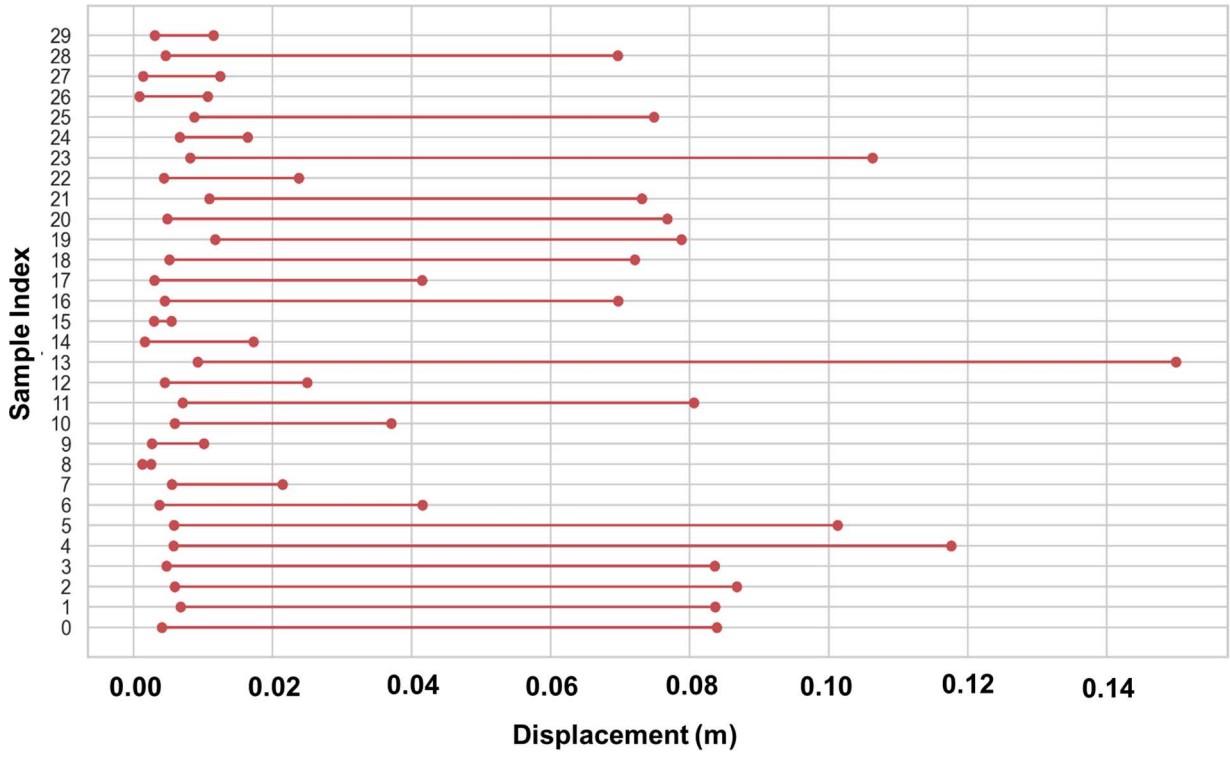

**Fig 12. Comparison of roof displacement and inter-story drift in timber structures.**

error magnitude, classification correctness, and positive prediction balance. Table 7 shows the numerical outcomes of error metrics.

The proposed GBRF-SCO model performed best with R² (0.988), MAE (0.54), and RMSE (0.20). The Enhanced Deep LSD model showed lower performance with R² (0.98), MAE (0.68), and RMSE (0.23). Overall, the proposed model gives more accurate and reliable results in Table 8. Fig 15 shows the Benchmark assessment of GBRF-SCO classification algorithms using quantitative performance metrics.

To reduce bias and have a fair comparison, all the ML models used hyperparameter settings taken directly from their original and peer reviewed studies. The only change made on the testing was the data set used for training. The data before and after counting, testing data, and tools to analyze the counting were the same. Such all-in-one testing method guarantees each model's performance relies only on its skill and not on anything else apart from the model. The hyperparameter configurations for all the benchmark models used in the comparative analysis were used as they were in their original and peer-existing research to make the fair and unbiased comparison. Enhanced Deep LSD model makes use of best line confidence thresholds, crack length filter, and CNN based features extractor. The SVM makes use of polynomial kernel and a regularization parameter $C=45$. AdaBoost makes use of a decision tree base estimator (max depth = 10), 180 estimators and learning rate of 1.0. The calculator uses a gbtree booster with $Î_c=0.1$, 450 estimators, Lambdas = 1.0 and $Ï«=0.01$ with subsampling ratios of 0.7. The Stacking ant neural network integrated multiple ANN base learners to a meta learner of logistic regression ($C=20$). The Gradient boosting NN made use of 25 boosting iterations with a learning rate of 0.2 and the weak learners was an ANN with two hidden layers with 128 neurons each. The proposed GBRF-SCO hyperparameters are fine-tuned by the SCO, which allows for an adaptive adjustment of the hyperparameters for better seismic performance estimation. All models in general were analysed under the same data preparation and experimental conditions so that every model could be objectively compared and their use could be re-produced.

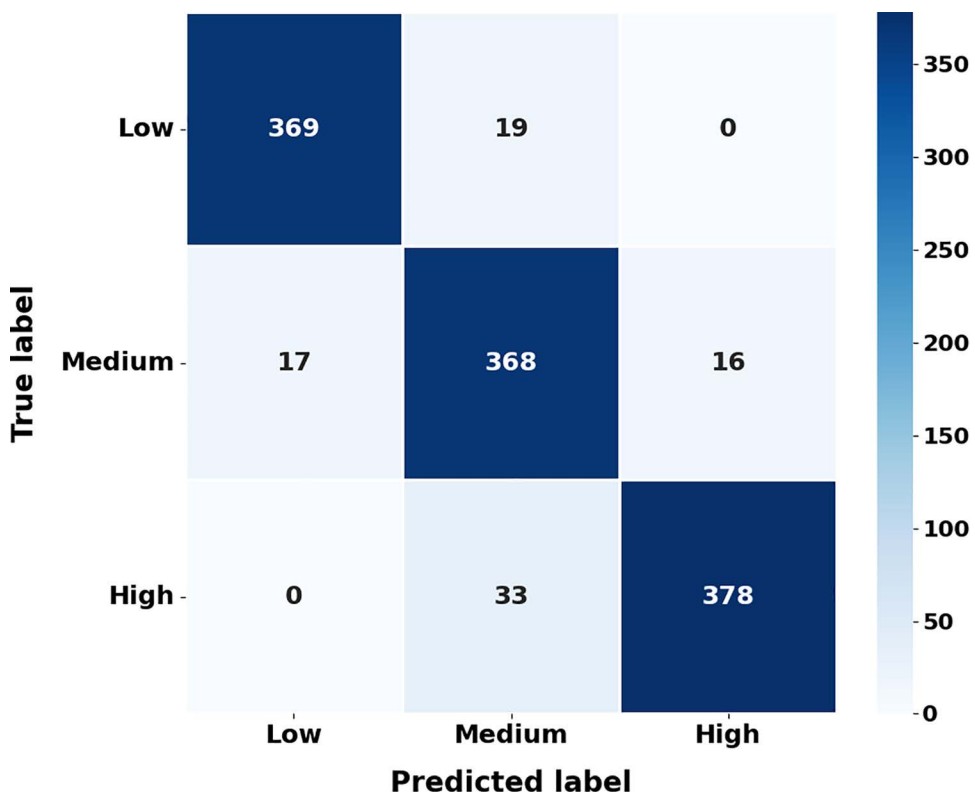

**Fig 13. Confusion matrix representation of roof displacementclassificarion results.**

The GBRF-SCO model outperformed all other methods, achieving the highest overall accuracy of 0.949, according to the performance comparison of several classification models. Along with the best ROC AUC Score (0.988), it had excellent performance across all other measures as well, including Precision (0.889), Recall (0.884), and F1-Score (0.897). The suggested model outperforms conventional models in terms of GBRF-SCO's predictive power and class separation efficiency. Table 9 shows the results of the ablation studies that compared the efficiency of the individual and combined models.

This ablation research showed that the combined GBRF-SCO model outperformed its individual components, GBRF and SCO. This integration led to higher accuracy, improved prediction stability, and better overall seismic performance optimization.

### 4.10. Discussion

The safety and resilience of wood buildings during earthquakes were enhanced through the development of the GBRF-SCO, capable of accurately forecasting and optimizing seismic performance parameters. Previous deep learning (DL) models for wood material identification showed promise but were dataset-specific and lacked generalizability [29]. Similarly, machine learning (ML) effectively assesses seismic damage in reinforced concrete (RC) structures, though its accuracy depends heavily on data quality and regional calibration [30]. Addressing these limitations, the study introduces the GBRF-SCO model, integrating AI-based prediction and optimization to improve seismic performance accuracy and design efficiency across diverse wood structure types.

**4.10.1. Practical implications.** Engineers can make safer, more resilient buildings by designing wood structures with excellent seismic properties using the GBRF-SCO framework. It offers a data-driven decision-support tool that real construction projects can use to make buildings more seismically compliant and safer.

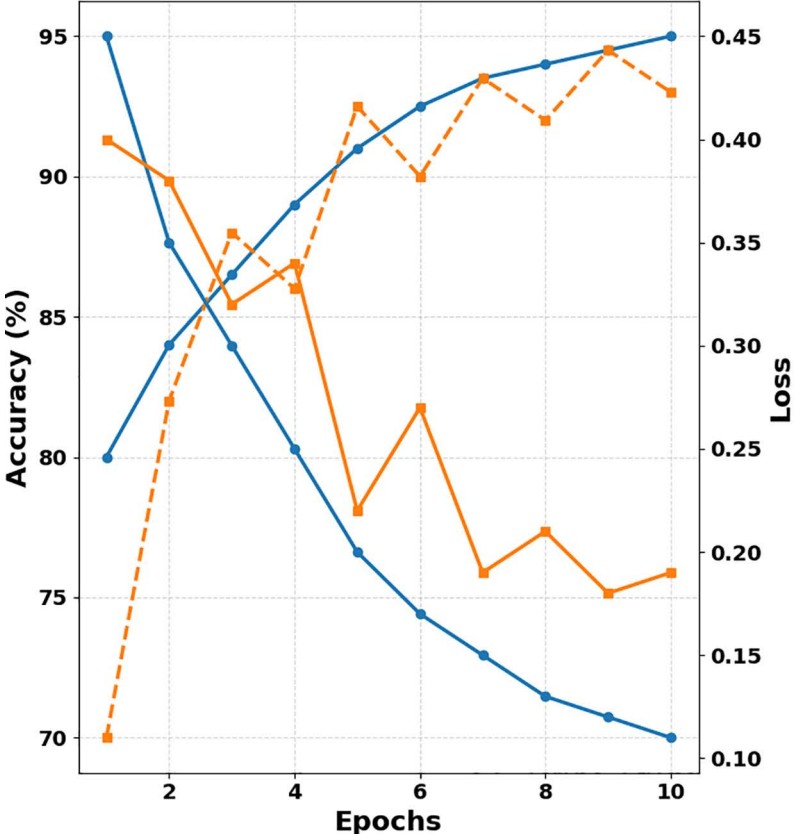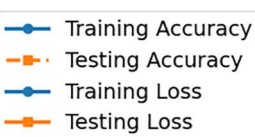

**Fig 14. Performance analysis of GBRF-SCO model across training epochs.**

**Table 7. Evaluation of predictive modelling approaches with error metrics.**

| Approaches | $R^2$ | MAE | RMSE |
|---|---|---|---|
| Enhanced deep LSD [29] | 0.98 | 0.68 | 0.23 |
| GBRF-SCO [Proposed model] | 0.988 | 0.54 | 0.20 |

**Table 8. Comparison of different classification model performances.**

| Approaches | Accuracy | Precision | Recall | F1-Score | ROC AUC Score |
|---|---|---|---|---|---|
| SVM [30] | 0.866 | 0.863 | 0.840 | 0.850 | 0.975 |
| AdaBoost [30] | 0.878 | 0.875 | 0.856 | 0.864 | 0.980 |
| XGBoost [30] | 0.882 | 0.874 | 0.863 | 0.868 | 0.982 |
| Stacking ANN [30] | 0.870 | 0.863 | 0.848 | 0.854 | 0.977 |
| Gradient boosting ANN [30] | 0.873 | 0.866 | 0.852 | 0.858 | 0.979 |
| GBRF-SCO [Proposed model] | 0.949 | 0.889 | 0.884 | 0.897 | 0.988 |

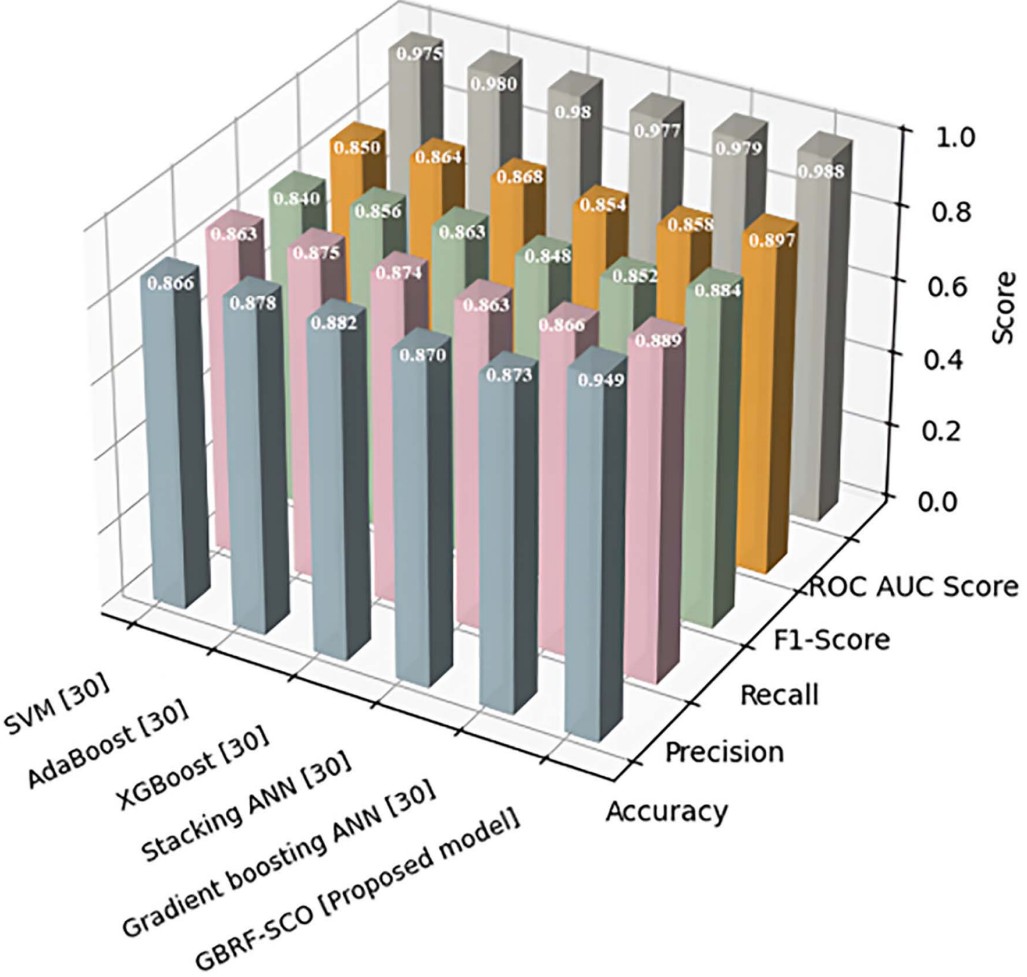

**Fig 15. Evaluation of classification model accuracy across multiple performance indicators.**

**Table 9. Ablation research results for model performance comparison.**

| Model | Precision | F1-Score | Accuracy | Recall |
|---|---|---|---|---|
| GBRF | 0.861 | 0.857 | 0.902 | 0.854 |
| SCO | 0.872 | 0.870 | 0.918 | 0.868 |
| GBRF-SCO (Proposed Model) | 0.889 | 0.897 | 0.949 | 0.884 |

## 5. Conclusion

An AI-assisted GBRF-SCO framework was proposed for accurately predicting and optimizing key seismic performance parameters of timber structures. By integrating GBRF-SCO, the framework effectively captures complex nonlinear structural-seismic relationships while adaptively optimizing model hyperparameters. The proposed approach demonstrates clear performance advantages over conventional machine learning and ensemble models, achieving an $R^2$ of 0.988, MAE of 0.54, and RMSE of 0.20. Beyond predictive accuracy, the GBRF-SCO framework provides practical

engineering value by supporting data-driven design decisions aimed at reducing inter-story drift and roof displacement, thereby enhancing the safety, resilience, and sustainability of timber buildings in seismic regions.

### 5.1. Limitations and future work

The GBRF-SCO framework has robust predictive and optimization capabilities; nevertheless, the study is limited by data diversity and availability, since the dataset predominantly depends on recorded and simulation-based data rather than comprehensive real-world experimental observations. The performance of the model depends on how good and complete the structural and seismic input parameters are. Also, the SCO-based optimization makes it more expensive to compute for high-dimensional feature spaces. Furthermore, extrapolation during intense seismic occurrences or atypical timber structural systems may be constrained by inadequate representative samples. Future efforts will focus on comprehensive experimental testing, field-based sensor data collection, multi-hazard scenario integration, and enhancements in computing efficiency to bolster resilience and practical application across various timber structures.

## Supporting information

**S1 File. We have uploaded the minimal dataset underlying the findings of this study as Supporting Information with the submission.** In addition, the dataset is publicly available without restriction from the Kaggle repository at the following link: https://www.kaggle.com/datasets/freshersstaff/timber-seismic-performance-dataset/data (XLSX)

## Author contributions

**Conceptualization:** Dongqi Wei.

**Data curation:** Feng Zhou.

**Investigation:** Dongqi Wei.

**Project administration:** Xuan Zhang.

**Writing – original draft:** Yuqiang Ding.

**Writing – review & editing:** Feng Zhou.

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
