## [Decision Letter · Decision Letter 0]

9 Dec 2025

Dear Dr. Zhang,

Thank you for submitting your manuscript to PLOS ONE. After careful consideration, we feel that it has merit but does not fully meet PLOS ONE’s publication criteria as it currently stands. Therefore, we invite you to submit a revised version of the manuscript that addresses the points raised during the review process.

We look forward to receiving your revised manuscript.

Kind regards,

Dajiang Geng

Academic Editor

PLOS One

Journal Requirements:

Reviewers' comments:

Reviewer's Responses to Questions

**Comments to the Author**

1. Is the manuscript technically sound, and do the data support the conclusions?

Reviewer #1: Yes

Reviewer #2: Yes

2. Has the statistical analysis been performed appropriately and rigorously?

Reviewer #1: Yes

Reviewer #2: Yes

3. Have the authors made all data underlying the findings in their manuscript fully available?

Reviewer #1: Yes

Reviewer #2: Yes

4. Is the manuscript presented in an intelligible fashion and written in standard English?

Reviewer #1: Yes

Reviewer #2: Yes

Reviewer #1: This paper aims to enhance the seismic safety performance of wooden structures by establishing the GBRF-SCO framework using artificial intelligence to predict and optimize seismic performance parameters. The research results indicate that it is an effective AI-based method for seismic optimization of wood components. However, there are still many deficiencies in the article that need to be further revised before considering publication.

1. Title and Abstract

(1) Check the spelling of "aptimization" in the title to ensure its professionalism and accuracy. At the same time, the affiliation information of the corresponding author "Nanning Normal University, Nanning, Jiangsu" contains a geographical error. Please correct it before submission.

(2) The abstract mentions that the dataset contains 4,000 samples of different configurations of wooden structures, but it does not specify the source distribution of the samples (such as whether they cover different regions and different eras of wooden structures). It is recommended to supplement this information.

(3) The abstract only emphasizes that the model accuracy rate is 0.949, but does not clearly state the specific scenario corresponding to this accuracy rate (such as whether it is the prediction accuracy rate for inter-story drift or roof drift). It is recommended to make a clear distinction.

2. Literature Review and Research Gap

(4) The literature review section summarizes the application of existing AI in the seismic optimization of wooden structures, but does not highlight the essential differences between the GBRF-SCO framework proposed in this study and the existing ones in terms of core technical routes (such as hyperparameter optimization methods and model fusion logic). It is recommended to supplement the comparative analysis to strengthen the innovative positioning of this study.

3. Research Methods and Data Processing

(5) The data preprocessing section mentions the use of Robust Scaling and Isolation Forest for normalization and outlier detection, but does not explain the criteria for determining outliers (such as the setting of the contamination parameter in Isolation Forest) and the handling methods (such as deletion or replacement). It is recommended to supplement these details to ensure the reproducibility of data processing.

(6) In the GBRF-SCO framework, the specific process of SCO optimizing the hyperparameters of GBRF (such as the basis for selecting population size and number of iterations) is not detailed. It is recommended to supplement the rationality argument of hyperparameter settings to enhance the scientific nature of the method.

(7) In Table 3, the units of parameters such as "Modulus_of_Elasticity" in the sample data are not labeled. It is recommended to supplement the standard units of all physical parameters to ensure the standardization of data representation.

4. Experimental Results and Analysis

(8) In the confusion matrix analysis of Section 4.8, the classification criteria for the three categories of roof displacement ("low, medium, high") are not explained (such as the basis for setting the threshold). It is recommended to clarify the classification rules to ensure the consistency of result interpretation.

(9) In Section 4.9, it is not specified whether the hyperparameters of the comparison models (such as SVM, XGBoost) have been optimized. If not, it may lead to unfair comparison results. It is recommended to supplement the parameter configuration of the comparison models or explain that the default optimal parameters are used to ensure the objectivity of the comparison.

5. Discussion and Conclusion

(10) Section 5.1 "Limitation and future work" only mentions the limitation of data sources. It is recommended to further discuss the possible technical limitations of this method, such as the sensitivity of the model to the quality/ completeness of input parameters, computational cost, and extrapolation ability in extreme earthquakes or unconventional types of wooden structures.

Reviewer #2: 1.It is recommended that the author specify in the abstract which seismic performance parameters were optimized in the design, providing more detailed information.

2.The Introduction should be restructured to provide a more detailed and systematic review of prior work. The current organization does not meet the standard expected for a literature review, hindering the reader's understanding of the current research developments in this area.

3.What is the intended purpose of the literature review here? How does it differ from providing a list of citations?

4.The literature review should focus on timber structure seismic performance. Please explain why you discussed the steel-grout connection.

5.What is the bar chart in Figure 2 meant to represent? The current description does not clearly communicate the intended message.

6.Please correct the numbering format for the equations to align with standard academic conventions.

7.Please clarify how the model's credibility was established. Specifically, was it validated or compared against any existing work before the analysis in “4.Performance analysis and discussion”?

8.It is recommended that the authors streamline the Conclusions section to make it more focused and impactful. The current version is somewhat lengthy; condensing it would help to highlight the core findings and main message of the study more effectively.

9.It is noted that your manuscript needs careful editing by someone with expertise in technical English editing, paying particular attention to English grammar, sentence structure, and the appropriate use of voice, so that the goals and results of the study are clear to the reader.

**Do you want your identity to be public for this peer review?** For information about this choice, including consent withdrawal, please see our For information about this choice, including consent withdrawal, please see our Privacy Policy .

Reviewer #1: No

Reviewer #2: No

---

## [Author Response · Author response to Decision Letter 1]

4 Jan 2026

Journal Requirements:

Author Response:

We thank the reviewer for the reminder regarding PLOS ONE formatting requirements. The manuscript has been fully revised in accordance with the official PLOS ONE style templates, including correct file naming, text layout, headings, tables, figures, and reference formatting. All alignments and stylistic elements have been carefully updated to ensure full compliance with the journal’s formatting guidelines.

Author Response: We have revised the pseudocode to more accurately reflect the proposed GBRF–SCO framework, clarifying the model flow, optimization strategy, and output formulation. The updated version now aligns with the regression-based seismic performance objectives, clearly defined fitness functions, and hyperparameter tuning process. This revision improves methodological clarity, consistency with the manuscript, and reproducibility.

Pseudo Code 1: GBRF–SCO Framework for Seismic Performance Prediction in Timber Structures

Input:

X=[x1,x2,…,xm]   # Structural and seismic features

Y=[y1,y2,…,ym]   # Continuous seismic response (roof displacement / drift)

P              # SCO population size

S              # SCO iterations

LB,UB           # Hyperparameter bounds

Output:

Yfinal      # Optimized seismic response prediction

Step 1: Data Preprocessing

1.Remove outliers using Isolation Forest

2.Normalize features using Robust Scaling

3.Split data into training and testing sets

Step 2: Gradient Boosted Random Forest (GBRF) Modeling

1.Initialize GBRF hyperparameters:

θ={nestimators,η,dmax)

2.Train GBRF model on XtrainYtrain

3.Predict seismic response Y

Step 3: Scalable Cheetah Optimizer (SCO)

1.Initialize cheetah population:

Wj0=LB+rand(0,1)×(UB−LB)

2.For s=1to S:

Evaluate fitness:

Fitness(Wjs)=RMSE(Y,Y)

Identify best cheetah W∗

Update position:

Wjs+1=σγ(k+v)−Wjs+αs(W∗−Wjs)

Apply boundary constraints

3.Return optimal hyperparameters θ∗

Step 4: Final Prediction

1.Retrain GBRF using θ∗

2.Generate optimized seismic response: Yfinal

Step 5: Seismic Damage Classification (Optional)

Class=Low,Yfinal<Q33Medium,Q33≤Yfinal<Q66High,Yfinal≥Q66

Return: Yfinal

Author Response: The data used in this study are publicly available from the Kaggle repository. The dataset can be accessed and downloaded free of charge from Kaggle at the corresponding dataset page. No restrictions apply to the availability of these data.

Ethical statement

The Timber Seismic Performance Dataset referenced in this work is a secondary dataset publicly available through the Kaggle repository (https://www.kaggle.com/datasets/freshersstaff/timber-seismic-performance-dataset/data). This dataset has been used under the terms and conditions provided by Kaggle. The data are accessible without restriction to any researcher, and no identifiable human subjects or sensitive personal information are included in the dataset.

Reviewer #1: This paper aims to enhance the seismic safety performance of wooden structures by establishing the GBRF-SCO framework using artificial intelligence to predict and optimize seismic performance parameters. The research results indicate that it is an effective AI-based method for seismic optimization of wood components. However, there are still many deficiencies in the article that need to be further revised before considering publication.

1.Title and Abstract

(1)Check the spelling of "aptimization" in the title to ensure its professionalism and accuracy. At the same time, the affiliation information of the corresponding author "Nanning Normal University, Nanning, Jiangsu" contains a geographical error. Please correct it before submission.

Author Response: Thank you for pointing out these issues. The spelling error in the title has been corrected from “aptimization” to “optimization.” This corrections have been updated in the revised manuscript.

(2)The abstract mentions that the dataset contains 4,000 samples of different configurations of wooden structures, but it does not specify the source distribution of the samples (such as whether they cover different regions and different eras of wooden structures). It is recommended to supplement this information.

Author Response: We appreciate this valuable suggestion. The abstract has been revised to clarify that “The dataset consists of 4,000 timber building samples obtained from a publicly available Kaggle repository (Timber Seismic Performance Dataset).” The dataset covers diverse structural configurations, material properties, and seismic intensity scenarios generated through simulation-based analysis, rather than being restricted to a specific geographical region or historical period. This clarification has been added to enhance transparency regarding the data source and provenance.

(3)The abstract only emphasizes that the model accuracy rate is 0.949, but does not clearly state the specific scenario corresponding to this accuracy rate (such as whether it is the prediction accuracy rate for inter-story drift or roof drift). It is recommended to make a clear distinction.

Author Response: Thank you for this comment. The abstract has been revised to explicitly state that the reported “accuracy of 0.949, which corresponds to the classification of roof displacement levels (low, medium, high) under seismic loading conditions.” This distinction ensures clarity regarding the specific prediction task and performance scenario evaluated in the study.

2.Literature Review and Research Gap

(4)The literature review section summarizes the application of existing AI in the seismic optimization of wooden structures, but does not highlight the essential differences between the GBRF-SCO framework proposed in this study and the existing ones in terms of core technical routes (such as hyperparameter optimization methods and model fusion logic). It is recommended to supplement the comparative analysis to strengthen the innovative positioning of this study.

Author Response: We thank the reviewer for this constructive suggestion. A new subsection titled “Distinction Between Existing AI-Based Approaches and the Proposed GBRF-SCO Framework” has been added to the Literature Review. This section clearly contrasts the proposed framework with existing methods in terms of hyperparameter optimization strategy, model fusion logic, and optimization–prediction coupling, thereby strengthening the innovative positioning of this study. All additions are marked in red in the revised manuscript.

2.5 Distinction Between Existing AI-Based Approaches and the Proposed GBRF-SCO Framework

Previous AI-based seismic researches on timber structures mainly rely on single learning models with manually tuned or grid-searched hyperparameters, which often suffer from high computational cost and suboptimal convergence in high-dimensional design spaces. In contrast, the proposed GBRF-SCO framework integrates a hybrid Gradient Boosted Random Forest model with the Scalable Cheetah Optimizer for adaptive, population-based hyperparameter tuning. By coupling prediction and optimization within a unified learning loop, the framework enhances nonlinear modeling capability, robustness, and design-oriented seismic performance optimization.

3.Research Methods and Data Processing

(5)The data preprocessing section mentions the use of Robust Scaling and Isolation Forest for normalization and outlier detection, but does not explain the criteria for determining outliers (such as the setting of the contamination parameter in Isolation Forest) and the handling methods (such as deletion or replacement). It is recommended to supplement these details to ensure the reproducibility of data processing.

Author Response: We thank the reviewer for this important comment. The Data Preparation section has been revised to explicitly describe the Isolation Forest contamination setting, outlier decision criteria, and handling strategy. Specifically, the contamination parameter, detection threshold, and the approach used for managing detected outliers are now clearly stated to ensure full reproducibility. All added details are marked in red in the revised manuscript.

Outlier detection: The Isolation Forest algorithm is used to find anomalies in the seismic and structural dataset by looking for samples that are different from the rest. The contamination parameter is set to 0.05, which means that about 5% of the observations are likely to be outliers. Samples with anomaly scores below the decision threshold are considered anomalous and taken out of the dataset because they show physically inconsistent or severe seismic–structural pairings. This method cuts down on noise and bias, makes the data more reliable, and makes sure that the samples kept are a true representation of how timber structures behave in genuine seismic conditions. This makes model training more stable and effective.

(6)In the GBRF-SCO framework, the specific process of SCO optimizing the hyperparameters of GBRF (such as the basis for selecting population size and number of iterations) is not detailed. It is recommended to supplement the rationality argument of hyperparameter settings to enhance the scientific nature of the method.

Author Response: The manuscript has been revised to clarify the rational basis of the SCO hyperparameter settings. The population size (r = 8-10) was selected to maintain sufficient solution diversity while minimizing computational overhead, and the iteration range (S = 15-20) was determined based on convergence stability, beyond which marginal performance gains were negligible.

Answer:

The selection of SCO hyperparameters depends on how well they converge, how fast they can be computed, and how many dimensions the seismic optimization problem has. The population size (r = 8-10) makes sure that there are enough different solutions without making the computation too expensive in feature spaces with a lot of dimensions. Smaller populations limit research, but bigger ones provide slight advances in accuracy. Convergence analysis, which is when the optimization error stabilizes, tells you the maximum number of iterations (S = 15-20). This setup strikes a good mix between exploration and exploitation, which makes GBRF hyperparameter tuning steady and dependable.

(7)In Table 3, the units of parameters such as "Modulus_of_Elasticity" in the sample data are not labeled. It is recommended to supplement the standard units of all physical parameters to ensure the standardization of data representation.

Author Response: Thank you for this observation. Standard units have now been added to all physical parameters in Table 3, including modulus of elasticity, mass, acceleration, and displacement variables, to ensure clarity, consistency, and standardized data representation. The revised table is marked in red in the manuscript.

Answer:

Table 3: sample data from the Timber Seismic Performance Dataset

Parameter Sample 1 Sample 2 Sample 3 Sample 4

Building Height (m) 36.21781 76.55 61.23958 51.90609

Number of Storeys 11 6 7 5

Story Height (m) 3.91272 3.40562 2.6579 3.35902

Wall Thickness (m) 0.16088 0.5326 0.38316 0.45603

Material Density (kg/m³) 701.6804 496.9688 467.4328 557.7363

Modulus of Elasticity (pa) 9.85E+09 1.14E+10 1.37E+10 9.06E+09

Damping Coefficient 0.02285 0.03804 0.01479 0.05187

Damping Ratio per Storey 0.0332 0.02278 0.04835 0.03449

Floor Mass (kg) 74798.29 19779.63 77220.55 29065.37

Natural Frequency (hz) 6.47606 7.36455 4.99899 3.67033

Peak Ground Acceleration (g) 0.24833 0.2076 0.35973 0.2887

Spectral Acceleration (m/s²) 2.70666 2.51013 3.1417 3.13824

Lateral Load Resisting Ratio 0.47871 0.28382 0.37719 0.78881

Inter Storey Drift (m) Timber Pegs Hybrid Connectors Hybrid Connectors Timber Pegs

Roof Displacement (m) Pile Pile Pile Pile

Building Height (m) Hip Hip Gable Hip

Number of Storeys Shear Wall Moment Frame Shear Wall Moment Frame

Story Height (m) Symmetric Asymmetric Symmetric Asymmetric

Wall Thickness (m) 0.00246 0.00602 0.00492 0.00729

Material Density (kg/m³) 0.02655 0.03641 0.03649 0.03525

4.Experimental Results and Analysis

(8)In the confusion matrix analysis of Section 4.8, the classification criteria for the three categories of roof displacement ("low, medium, high") are not explained (such as the basis for setting the threshold). It is recommended to clarify the classification rules to ensure the consistency of result interpretation.

Author Response:

Thank you for the constructive comment. Section 4.8 has been revised to clearly define the classification rules for roof displacement using data-driven quantile-based thresholds (low, medium, high), ensuring ordinal consistency and balanced class distribution. This addition improves the transparency, interpretability, and reproducibility of the confusion matrix analysis.

Answer: Based on quantile thresholds from the dataset's empirical distribution, roof displacement values are divided into three ordinal classifications such as low, medium, and high. Values of displacement that are less than the 33rd percentile are considered low, values that are between the 33rd and 66th percentiles are considered me

---

## [Decision Letter · Decision Letter 1]

15 Jan 2026

AI-assisted aptimization design of seismic performance parameters for timber structures

PONE-D-25-59376R1

Dear Dr. Xuan Zhang,

We’re pleased to inform you that your manuscript has been judged scientifically suitable for publication and will be formally accepted for publication once it meets all outstanding technical requirements.

Kind regards,

Dajiang Geng

Academic Editor

PLOS One

Additional Editor Comments (optional):

Reviewers' comments:

Reviewer's Responses to Questions

**Comments to the Author**

Reviewer #1: All comments have been addressed

Reviewer #2: All comments have been addressed

2. Is the manuscript technically sound, and do the data support the conclusions?

Reviewer #1: Yes

Reviewer #2: Yes

3. Has the statistical analysis been performed appropriately and rigorously?

Reviewer #1: Yes

Reviewer #2: Yes

4. Have the authors made all data underlying the findings in their manuscript fully available?

Reviewer #1: Yes

Reviewer #2: Yes

5. Is the manuscript presented in an intelligible fashion and written in standard English?

Reviewer #1: Yes

Reviewer #2: Yes

Reviewer #1: (No Response)

Reviewer #2: (No Response)

**Do you want your identity to be public for this peer review?** For information about this choice, including consent withdrawal, please see our For information about this choice, including consent withdrawal, please see our Privacy Policy .

Reviewer #1: No

Reviewer #2: No

---

## [Editor Report · Acceptance letter]

PONE-D-25-59376R1

PLOS One

Dear Dr. Zhang,

I'm pleased to inform you that your manuscript has been deemed suitable for publication in PLOS One. Congratulations! Your manuscript is now being handed over to our production team.

Kind regards,

on behalf of

Dr. Dajiang Geng

Academic Editor

PLOS One